# Public engagement for inclusive and sustainable governance of climate interventions

Livia Fritz ⬵[1] ✉, Chad M. Baum ⬵[1], Sean Low ⬵[1] & Benjamin K. Sovacool[1,2,3]

The need for public engagement is increasingly evident as discussions intensify around emerging methods for carbon dioxide removal and controversial proposals around solar geoengineering. Based on 44 focus groups in 22 countries across the Global North and Global South (N = 323 participants), this article traces public preferences for a variety of bottom-up and top-down engagement practices ranging from information recipient to broad decision authority. Here, we show that engagement practices need to be responsive to local political cultures and socio-technical environments, while attending to the global dimensions and interconnectedness of the issues at stake. Establishing public engagement as a cornerstone of inclusive and sustainable governance of climate-intervention technologies requires (i) recognizing the diversity of forms and intensities of engaging, (ii) considering national contexts and modes of engagement, (iii) tailoring to technological idiosyncrasies, (iv) adopting power-sensitive practices, (v) accounting for publics' prior experience, (vi) establishing trust and procedural legitimacy and (vii) engaging with tensions and value disagreements.

In post-Paris climate assessments and governance, scientists, policymakers and businesses around the world have been devoting growing attention to emerging climate-intervention technologies, in the past often collectively referred to as climate (geo)engineering. These include approaches to carbon dioxide removal (CDR) – a diverse, burgeoning range of methods for absorbing carbon dioxide from the air – as well as more controversial proposals around solar radiation modification (SRM) for offsetting global warming by reflecting incoming sunlight. Decisions on which approaches to pursue at large scale and how to govern them are only in formation, and wider public debates are nascent at best.

In response to past controversies about science and technology, many observers and scholars have called for a "participatory turn"[1–3], a "participatory return"[4] or a "deliberative turn"[5], "upstream" public engagement[6] and the opening-up of assessments through "societal appraisal"[7]. When it comes to novel technologies and the societal transformation pathways they condition, such involvement becomes even more pertinent.

Experiences in adjacent fields such as the social acceptance of energy systems have shown how the lack of meaningful public engagement can provoke controversy, for example, over shale gas[8] and carbon capture and storage (CCS)[9], or conversely, have pointed to the significance of public engagement and energy citizenship for implementing the energy transition[10,11]. Under the umbrella of responsible research and innovation (RRI), there is widespread recognition that public values and interests need to be considered in the governance of emerging technologies[12–17]. Public and stakeholder engagement is considered especially important in new sectors in which legal regulations are not yet clearly defined or routinized decision-making processes are not yet in place[18], as well as when stakes are high, values disputed, and uncertainties abound[19].

[1]Department of Business Development and Technology, Aarhus University, Birk Centerpark 15, 7400 Herning, Denmark. [2]Science Policy Research Unit (SPRU), University of Sussex Business School, Jubilee Building, Arts Rd, Falmer, Brighton BN1 9SL, UK. [3]Department of Earth and Environment, Boston University, 685 Commonwealth Ave, Boston, MA 02215, USA. ✉e-mail: livia.fritz@btech.au.dk

In this context, calls have multiplied for involving publics in the governance of geoengineering experiments[20,21] and science policy decision-making more generally[22–26]. Public participation figures in the so-called Oxford Principles – a set of norms that have been put forward as good practice for governing geoengineering research and innovation[25] and that some deem a requirement before even considering geoengineering a legitimate object of governance[20,27].

At the same time, the lack of adequate public involvement and resulting contestations and controversies have contributed to the termination or suspension of tests and experiments, notably in the case of the Harvard-led SCoPEx program, giving further rise to calls for meaningful societal appraisal[28]. Also in the context of CDR, previous experiences with afforestation or restoration of ecosystems, such as peatlands, point to the importance of public participation and the inclusion of local actors and communities[29–31]. Public engagement and community involvement are also called for regarding bioenergy with carbon capture and storage (BECCS)[32,33], in light of past controversies around the negative impacts of its bioenergy component on local communities[34]. Public engagement is also increasingly discussed with regard to more nascent ocean-based CDR methods such as ocean alkalinity enhancement[35–37].

Public engagement is, consequently, essential for assessing the desirability and feasibility of emerging climate-intervention technologies. Researchers, particularly from the social sciences, have responded to these calls and created deliberative spaces via focus groups, group facilitations and workshops in order to elicit public hopes and concerns regarding selected climate-intervention technologies[38] and render visible the value systems, local experiences and socio-political contexts in which these emerge.

Deliberative exercises have provided us with insights into public perceptions of climate-intervention technologies in the making. They have revealed public preferences for approaches that are perceived as "natural"[39] and have identified controversy spill-overs and public concerns surrounding geological storage[40,41]. They have uncovered concerns over unsustainable land use practices[42], and over delays in emissions reduction and deep, system-wide transformations[43,44]. These deliberation experiences also suggest that publics – unlike their role postulated in "deficit" models[45] – are capable of discussing and making sense of technological issues with limited prior knowledge[46]. Recent research, furthermore, shows that publics in the Global South and Global North demand further information and engagement campaigns about CDR and SRM approaches[47].

From the perspective of public engagement, participation and deliberation, important methodological and procedural questions remain – particularly when considering the scale at which climate-intervention technologies would need to be deployed. Little is known about which forms of engagement are meaningful for different climate-intervention approaches and the socio-political contexts in which they are situated or the publics that form around them. Frameworks for adequate participation might differ across "governance frameworks, power inequities, and opposing perspectives of what consists of 'good technology' or 'good knowledge'"[48].

In this paper, relying on a large qualitative dataset from 44 focus groups in 22 countries (Fig. 1), we ask: what do diverse publics think about their own role when it comes to decision-making about CDR and SRM approaches? Our aim is to trace (i) how publics in diverse socio-political settings would like to engage in governing and/or implementing the respective climate-intervention technologies, (ii) why they consider different forms of engagement desirable and (iii) which conditions they identify as crucial for meaningful engagement with climate-intervention technologies.

We build on different participation theories and previous deliberative work, consisting mostly of discrete, western-centered participation events, and argue that moving toward a systemic approach to public engagement is required[49] – one that attends to diverse and situated conceptions of who the public is and how it can contribute to just and sustainable governance arrangements for emerging climate-intervention technologies. Such a shift in perspective allows us to move beyond institutional framings of publics and participation procedures[33] and to draw an empirically rich picture of diverse,

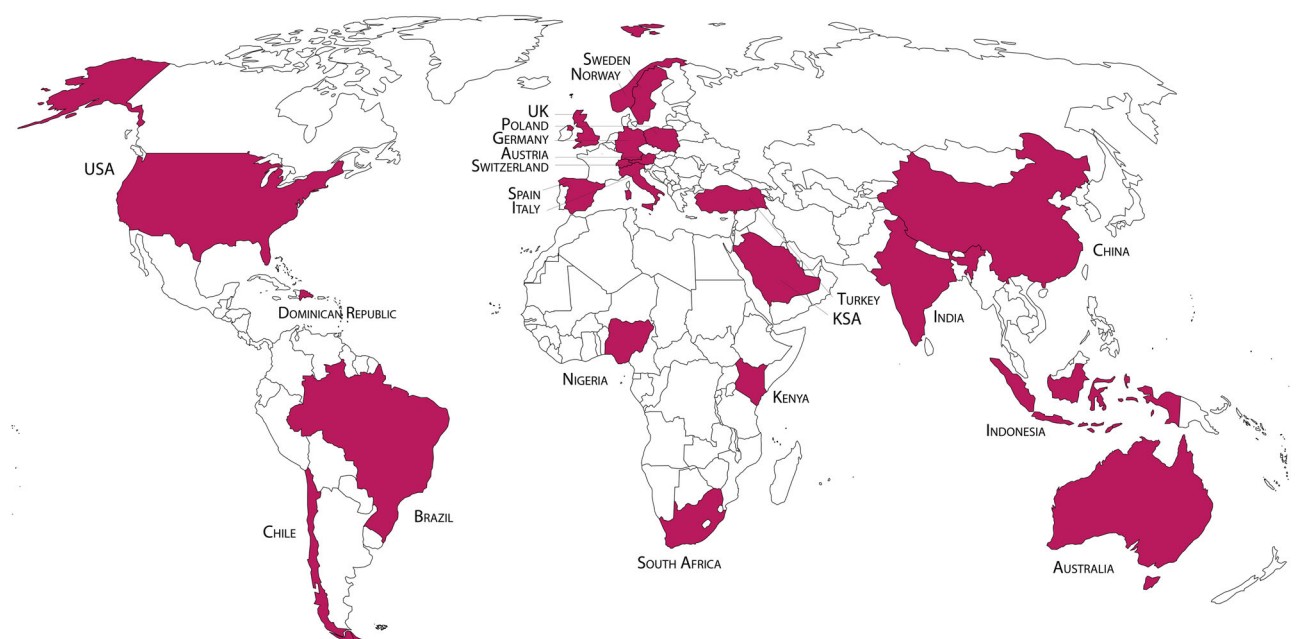

**Fig. 1 | Overview of 22 countries included in the sample of focus groups (N = 323 total participants).** This paper is based on the qualitative analysis of 44 focus groups conducted in 22 countries (with one focus group in each country in a rural and one in an urban environment), involving in total 323 participants with diverse socio-demographic backgrounds. Following the classification of United Nations' Finance Center for South–South Cooperation the countries covered in this paper include 10 countries from the Global South: Brazil, Chile, India, Indonesia, South Africa, Kenya, Saudi Arabia, Nigeria, Dominican Republic, China and 12 countries from the Global North: USA, Australia, Austria, Germany, UK, Sweden, Poland, Switzerland, Italy, Norway, Spain and Turkey.

bottom–up framings of publics and modes of participation that vary across socio-political contexts. Such framings are, moreover, sensitive to the technologies in question; they are embedded in conceptions of who and what knowledge can be trusted, and what values matter. On this basis, we outline conditions for meaningful public engagement as a cornerstone of inclusive and sustainable governance of CDR and SRM approaches.

## Results

### Conceptualizing public engagement

Participation and engagement are terms with many forms, functions and objectives, and been applied in the assessment of science and technology, climate change, energy and sustainability. In this section, we provide an overview of conceptions of participation according to the underlying motivations, as well as according to the degree of involvement, and highlight the systemic nature of participation practices (Table 1). We illustrate how different understandings and theories of participation come with specific roles for publics, and influence which formats of engagement are considered meaningful.

A first branch of scholarship conceptualizes participation and engagement by motivation. Environment and sustainability-related debates in this tradition revolve around whether participation processes are or should be implemented for functional reasons, or because democratization is seen as a key ingredient for transformations toward just and low-carbon societies[50,51].

Laying the ground for such conceptualizations, Fiorino[52] identified three imperatives: the normative imperative, in which participation is an end in and of itself; the substantial imperative, in which participation has the goal of achieving better ends; and the instrumental imperative, in which participation is aimed at securing specific interests. The first is process-oriented, while the latter two focus on outcomes[7]. The goal of outcome-oriented approaches can be to improve decision-making by including diverse forms of knowledge, values, priorities and visions; and to build trust, foster ownership and encourage collective learning[53]. A recent meta-analysis[54] finds a positive effect of public participation on the outcomes of environmental governance, but cautions that effects depend on the decision-making context and the goals of the environmental agencies involved.

Calls for participation for instrumental reasons are also commonly found in energy and environmental planning – for example, as a response to local opposition and aimed at securing buy-in. In the latter, publics have frequently been conceptualized as motivated by self-interest and opposition has been reduced to and framed as "NIMBY-ism" (Not-In-My-Back-Yard) – despite an increasing body of evidence putting these simplified conceptions in question[55–57].

Similarly, calls for engaging the public early in the development of novel technologies and the planning of their implementation are often underpinned by instrumentalist rationales and motivated by the aim of ultimately increasing public acceptance. Scholars from the field of Science and Technology Studies (STS) have shown how in many cases experts have implicitly framed publics based on a "deficit model", suggesting that publics contest complex issues because of a lack of knowledge, and that hence better science communication and education is needed to reduce skepticism toward science and technology[45,58].

Examples of such framings abound in science- and technology-related controversies about genetically modified organism techniques[59], nuclear energy[60] nanotechnology[61] or shale gas extraction[62] and more recently renewable energy technologies[63]. Waller et al.[33] argue that despite growing awareness and acknowledgment of the limitations of the deficit model of public understanding of science, such conceptions also persist in emerging literature on public engagement with CDR methods, which strongly focuses on questions of public acceptability and highlights low familiarity of publics with these approaches.

The participation literature tends to focus on how experts construct publics. Eaton et al.[64] trace how energy experts imagine publics for the case of bioenergy, differentiating broadly between "resisting" vs. "accepting" publics, as well as "active" vs. "passive publics". With a few exceptions such as Michael's[65] differentiation between "publics in general" and "publics in particular", publics' conceptions of their own roles and ways of engaging with science and technology are rarely elicited[63].

A second branch of scholarship differentiates types of participation regarding the degree to which publics are involved in a planning, design, decision-making or knowledge-production process.

Participation is frequently conceptualized in reference to the "Ladder of Participation"[66] that Arnstein developed for community planning. She differentiates eight uses of participation according to the degree of societal involvement, broadly distinguishing between non-participation, tokenism and citizen power. Adaptions of the ladder for climate, energy and sustainability research are numerous[67–69], using degrees of involvement to differentiate, for example, between information, consultation, cooperation, collaboration and empowerment[68]. Thinking about the role of society and publics in terms of these levels of engagement has also been proposed for solar geoengineering[21].

Arnstein's conceptualization implies that "the more participation, the better", thereby imagining publics as pre-existing entities waiting to be mobilized via top–down-initiated processes. Moving up the ladder of participation comes with the redistribution of power from the power-holding actors or institutions to the to-be-empowered citizens[2]. Participation is, thus, seen as way of challenging power relations and the societal status quo[70]. In this tradition, many advocates of participation pursue Habermas'[71] principles of ideal speech and assume that social interactions are "[…] competent and free from delusion, deception, power and strategy"[72]. Analysts and practitioners in this tradition tend to focus on consensus rather than on tensions and conflicts amongst participants[50,73]. Given fundamental differences in perspectives and at times incommensurable values the quest for consensus has, however, been called into question in the case of sustainability issues and transformation pathways[74].

Ladder-based conceptions of participation have, furthermore, been criticized for their focus on institutional framings of participation[33], an emphasis of "invited" spaces of participation[75] as well as discrete forms of engagement[2], with limited attention given to

**Table 1 | Participation and public engagement: overview of typologies and approaches**

| Approach | Key concepts | Illustrative publications |
|---|---|---|
| Typologies by rationale | Differentiate types of participation regarding the motivations that drive them, including instrumental, normative and substantial motivations | 50,52,53 |
| Typologies by degree or intensity | Differentiate types of participation regarding the degree to which publics are involved in planning, design, decision-making or even knowledge-production process; degrees include, for example, information, consultation, cooperation, collaboration, empowerment | 21,66–69 |
| Relational and systemic approaches | Analyse the embedding of participation practices in wider socio-political systems and considers emerging and bottom–up participation practices; "ecologies of participation" | 49,76,77,85 |

"claimed" spaces[75], bottom–up and emerging forms of engagement[76]. It has been shown on the example of the energy system how such narrow and fragmented perspectives on public engagement have failed "to capture the diverse, multiple and interconnected ways in which publics engage with energy systems on an ongoing basis"[49]. Normative theories of participation have neglected public preferences for participation, leaving questions about whether, how and when publics want to participate in climate governance or the shaping of novel technologies unanswered[51].

Public participation scholars in the field of climate, energy and sustainability have increasingly been attentive to the ways in which processes of public participation are enmeshed with wider systems and practices and embedded in socio-material settings and socio-political cultures[2,77,78].

Inspired by deliberative democracy, practice theory and STS, systemic perspectives that move beyond institutional, top–down forms of participation and individual participation processes or deliberation exercises are gaining traction. To grapple with the systemic nature of participation, Chilvers et al.[76] look at "ecologies of participation". Following this approach, participation practices comprise three main elements: subjects (participating actors), objects (issues and concerns) and models of participation (procedural formats). With the ambition of translating such a perspective into empirical observations, Chilvers et al.[49] propose to map public engagement with energy systems on two spectrums: between institution-led and citizen-led, and from issue formation to action.

Such a broader perspective enables us to attend to "the diverse, ongoing, already existing practices and settings through which publics are engaging with energy system transitions in different ways"[49], thereby also constructing publics in an active role, as agents engaging in their respective capacities and contexts rather than passive entities waiting to be informed or mobilized in a top–down fashion. Similarly, in the context of sustainability and transformation research, movements from deficit to co-production models treat publics as holders of diverse knowledges needed for addressing contemporary crises[79].

Our framework for qualitatively analyzing the focus groups was informed by these systemic approaches to participation, and typologies by degree of involvement and rationales (see "Methods"). Our key analytical dimensions on the "how", "what", "why" and "who" of public engagement with climate-intervention technologies structure the presentation of results.

We first present the different ways participants in the focus groups describe how publics should engage with climate-intervention technologies ("how"). Second, we show how preferences for different forms of engagement vary depending on the specific CDR and SRM approach in question ("what"). Third, we identify the rationales and motivations underpinning the forms of engagement they consider desirable or necessary, as well as trace arguments of why public engagement is not considered germane in some cases ("why"). Finally, we show which conceptions of the public accompany participants' respective views ("who") – rather than forming its own section, this theme is distributed throughout the results.

## How to engage
In this section, we report on the forms and intensities of public engagement. Members of the public in 44 focus groups speak in nuanced terms about their own role and ways of engaging with climate-intervention technologies. They describe a variety of forms of public engagement with CDR and SRM approaches, covering a wide spectrum of roles for publics ranging from passive recipients of information to active decision-makers (Fig. 2A). Mapped along the dimensions suggested by Chilvers et al.[49] a rich picture of potential formats of engagement with climate-intervention technologies emerges between top–down, institution-led approaches and citizen-led, bottom–up approaches (Fig. 2B). Participants' narratives and conceptions of how publics should engage with CDR and SRM vary across countries and technologies (Table 2).

Information and education: the most discussed form of engagement refers to receiving information and education relevant for making sense of CDR and SRM approaches. The desire for being better informed is frequently mentioned in Global North and Global South focus groups, confirming representative survey results[47]. Discussed in relation to all climate-intervention approaches, here publics attributed to themselves a passive role as recipients of information in the form of leaflets, information packages, talks or workshops prepared by governmental agencies, educational institutions like schools and universities, and shared via traditional media, social media and personal communication channels as well as promoted by celebrities. However, in most narrations, information and education only constitutes a first step, and pave the way for more intense forms of engagement.

In some cases, participants also see themselves as educators or at least information spreaders, sharing their knowledge about climate-intervention technologies and climate change in general with personal networks – a role that incorporates more active conceptions of the public. In such narrations, specific publics appear as brokers or translators of knowledge:

"I think I should go back to my community. Because sometimes you don't understand your topic, you have to decode your topic first in your language, that is the only way you can do better, so I think I have to go back to my community, educate the farmers in the best language that they will understand." (Nigeria urban)

Self- and community engagement: the second-most discussed form of engagement concerns practical ways in which participants envision contributing to realizing specific approaches. When talking about this form of engagement, participants ascribe an active role to publics and conceive them as agents of change – for example, in tree planting, ecosystem restoration activities, or implementing soil carbon approaches. In the lattermost case, participants also speak of specific publics such as farmers. The following reflection on supporting the implementation of afforestation efforts illustrates such self-engagement:

"This is the cheapest, effective, tried, and tested way of dealing with the carbon dioxide removal. But I feel that, the common man and the government should be involved on this. Whereby I'm being included so that I can participate myself, even without the government, I can individually participate in growing of these trees… planting all these trees." (Kenya urban)

Further formats of self-engagement include volunteering, financial donations, and participation in crowdfunding campaigns. For the engineered CDR methods – particularly DACCS and BECCS – which tend to be perceived as capital-intense, some participants mention indirect financial engagements via taxation. Regarding soil carbon approaches, this participant in a Saudi Arabian focus groups considers bottom–up or community public financing:

"Since these procedures will be highly expensive for sure, we as citizens can support these projects financially or in many different ways; […] through purchase or donations, government sectors or private sectors could have donation boxes for fundraising, we have to play any role." (Saudi Arabia urban)

While addressed in a plurality of groups across Global North and Global South countries, the emphasis of self-engagement is comparatively stronger in focus groups in the Global South. It is for biogenic approaches like afforestation and restoration of vegetation and soil carbon capture that participants feel they can most concretely

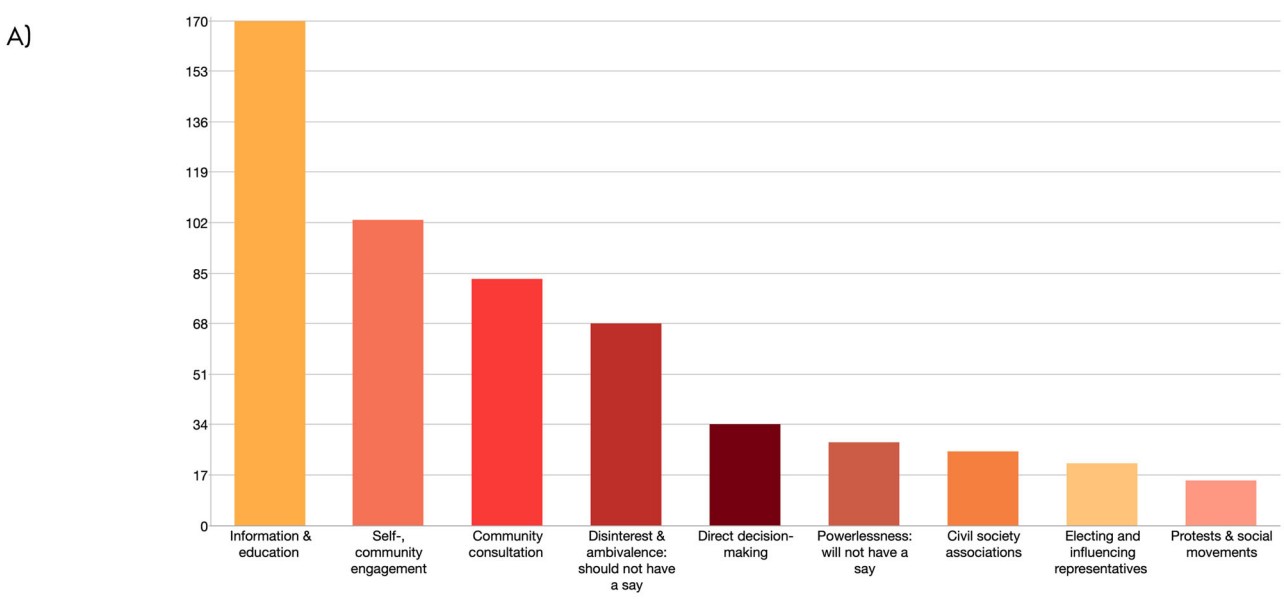

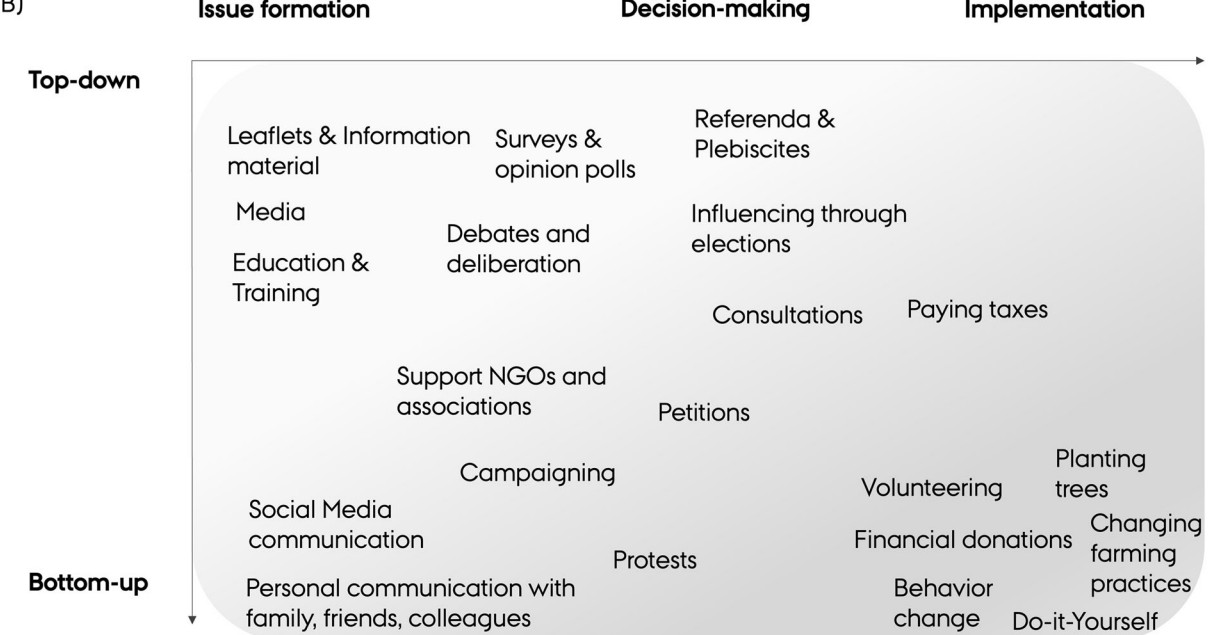

**Fig. 2 | Forms and formats of public engagement with climate-intervention technologies mentioned in focus group discussions in 22 countries.** (**A**) shows forms of public engagement and is based on the number of coded segments for the respective engagement categories; categories are not mutually exclusive; different colors signal different forms of engagement; (**B**) breaks these broader engagement categories into more specific formats & methods discussed by participants. Using a mapping grid adapted from Chilvers et al.[49], it shows whether they are rather top–down or bottom–up initiated and whether they occur in initial issue formation (information; expression of views), decision-making or implementation of climate-intervention technologies; categories not mutually exclusive.

contribute themselves. Notably, these various forms of active self-engagement are considerably less mentioned in relation to engineered CDR methods such as DACCS and even less for the SRM approaches. This differentiated approach by technology group is reflected in the following statement made in the context of discussions around SRM governance:

"I do believe this is not a one solution fits all; if it's a technical approach for strategies like this, let it be for the scientists that are going to come up with the solution. And for us as the primary members, we should be saying what we will be doing to play our part, that could be planting a tree. Obviously every effort helps." (South Africa rural)

Talking about self-engagement, participants repeatedly move beyond the immediate context of climate-intervention technologies and share their individual efforts in tackling climate change more broadly, including various examples of pro-environmental behavior changes. In some narrations, a collective dimension manifests, for example when participants link their engagement to community efforts or to forms of social organization such as associations.

**Table 2 | Summary of forms of public engagement mentioned across focus groups and technologies**

| Form/Intensity | Methods & formats | Technological focus | Frequency across countries |
|---|---|---|---|
| Information | Leaflets, media, social media, education and training | All | High |
| Self- and community engagement | Volunteering, financial donations, changing farming practices, help planting trees, pro-environmental behavior changes | AF/RF, SOILS, CDR in general; | High |
| Community consultation | Townhall meetings; debates and deliberation surveys and opinion polls | CDR in general, DACCS, BECCS, and MCB, but emphasis less pronounced | High |
| Indirect influence on decision-making | Electing political representatives | SAI and SRM in general | Medium |
| Direct action and protests | Demonstrations; joining social movements; campaigning | SRM in general, but emphasis less pronounced | Medium |
| Civil society associations | Supporting or joining NGOs, associations, advocacy groups | CDR in general, but emphasis less pronounced | Medium |
| Direct decision-making | Referenda, plebiscite, petitions | SAI | High |
| Ambivalence and powerlessness | – | SRM in general and SAI | High |

Low frequency = mentioned in 7 countries or less; medium frequency = mentioned in 8–14 countries; high frequency = mentioned in 15 countries or more; "SRM in general" and "CDR in general" are used where participants refer collectively to SRM or CDR, without specifying which particular method or technology they speak to. This coding category was used in addition to approach-specific codes.
SRM solar radiation modification, CDR carbon dioxide removal, AF/RF afforestation,reforestation and restoration, SOILS soil carbon sequestration and biochar, DACCS direct air capture and carbon storage, BECCS bioenergy with carbon capture and storage, MCB marine cloud brightening, SAI stratospheric aerosol injection.

Community consultation: consultation processes that allow publics to express their preferences, provide feedback and voice concerns are the third most discussed form of engagement, voiced by groups across Global North and South countries. Consultations are mentioned for all approaches, with the highest frequency for CDR in general, DACCS and MCB. They are discussed in the context of scaling-up biogenic CDR methods as well in relation to siting of more engineered approaches.

A wide range of formats for consultation are suggested, including townhall meetings, debates and deliberative workshops, online surveys and opinion polls, with some discussion on whether existing governance frameworks (e.g., planning regulations) offer adequate consultation mechanisms for the context of climate-intervention technologies or whether innovation in governance and regulatory systems would be needed (e.g., in Germany).

Discussing BECCS and siting decisions, for example, a participant in an Australian focus group calls for consultation of those affected by a project:

"Look, I think if it's going to go up in your community, you need to have genuine consultation on it. Like, I'd be pretty upset. […]. So I think there's probably different levels of involvement and authority, but I do think that, yeah, if it is going to be in your area, they do need to consult the individuals in that particular community, or I think they would get a lot of people offside very quickly." (Australia rural)

Deliberations are also brought forward as valuable processes in themselves; not necessarily with a specific decision or outcome in mind. In the context of soil carbon sequestration, for example, one participant applauds the value of debate and of engaging with other points of view:

"Debates are always good because they open your point of view, even if you don't agree, they open new paths. When you promote the debate amongst people, in small groups, this has a positive impact." (Brazil rural)

Yet other calls for consultation are embedded in discussions about the financial costs of implementing CDR, concerns over financial burdens being placed on publics and general reflections about priorities in public spending. Here participants speak about the wider public or publics in general rather than about specific, locally affected, publics, as the following extract from a Kenyan focus group illustrates regarding DACCS:

"I go out of line on this, the air capturing with carbon storage. […]. It's a very high cost. Therefore, the person to be consulted is a taxpayer in general. Because I don't expect my government to come in and impose the technology whereby they're capturing the carbon, and using a bigger percentage of the taxpayers' money to impose this technology - I'm a horticultural farmer. I don't expect the government to come and tell me we are building this technology, yet we don't have roads to transport whatever I have from my farm. […] For you to come up with the technology that I'm not sure whether it was going to benefit me for now or not. So, I think on this the taxpayer should be consulted or negotiated with." (Kenya urban)

Lastly, reflections on consultative approaches are tied to skepticism toward or rejection of approaches that would transfer full decision-making authority to publics. In these cases, participants stress the non-binding character of many consultation processes. In other cases, they mirror the limited role of publics in non-democratic political systems:

"Participation is necessary, but we can only put forward some good suggestions. In fact, we can't really make decisions as ordinary people." (China rural)

Direct decision-making: direct decision-making is discussed as form of engagement in the majority of countries, with a stronger plurality in Global North countries. Publics are conceived of as political actors who exert decision authority in the governance of climate-intervention technologies. While being mentioned for all approaches, direct decision-making via referenda, plebiscites (in Chile) or petitions (in Switzerland) is discussed most with regard to SAI.

Given the global impact of SRM technologies, participants argue that those affected should be involved. In so doing, the sphere of who is being affected – and who should be directly involved in decision-making – is extended from local communities to the global population. As one participant from Turkey argues:

"I think that since it is an issue that concerns all humanity [...], and even the universe, everyone's opinion should be sought, from the smallest to the largest. Votes can be done, something like a poll can be done." (Turkey urban)

At the same time explicit expressions of the idea that publics "should not" or "will not" have a say are also most frequent for SRM approaches, suggesting particularly controversial views on the public's role in SRM governance (see section on Rationales against engagement).

Electing and influencing political representatives: electing and influencing representatives is mentioned in less than half of the countries and emphasized most in European and Anglo-Saxon countries. Mostly discussed in relation to SRM approaches, publics are here addressed in their role as citizens exercising their voting rights.

References to this form of engagement are tied to reflections on the role of citizens in representative democratic systems. While mostly emphasized by participants in western democracies, it is also mentioned aspirationally by participants – for example, in Turkey. Discussions about delegating authority to political representatives to take what is perceived to be the right decisions regarding climate-intervention technologies are mostly intertwined with arguments of why placing direct decision-making authority in the hands of publics is not needed or not desired, and reflect a generally high trust in institutionalized forms of politics and the political system of the respective countries. This form of engagement is, furthermore, based on the assumption that there is some level of public and political discourse about issues related to climate-intervention technologies. Similar reflections on delegating decision-making to political leaders can also be found in the focus groups in countries with non-democratic political systems like China and Saudi Arabia where there is ostensibly high trust in government. While the notion of indirect influence is less present here, the idea of delegation cuts across the spectrum of democratic and authoritarian systems.

Direct action, protests and social movements: protesting and joining social movements as a way of engaging with emerging climate-intervention technologies is addressed in half of the countries, with the strongest plurality in Anglo-Saxon, European and Latin American focus groups. This builds on literatures all suggesting the salience of civil disobedience and social organization as leverage points for climate action[80,81].

In the case of the UK, for example, numerous references to current protest movements and climate activism such as "Just Stop Oil" or "Extinction Rebellion" suggest that this form of engagement was particularly salient in the UK context when focus groups were conducted. As a form of claimed spaces of participation, protesting is discussed most vehemently for the SRM approaches, pointing to their particularly controversial nature. Publics are conceived as (pro-)active subjects, taking action to make their voices heard, even if institutionalized politics does not invite them to do so. The following extract from a discussion in an Australian focus group regarding SAI highlights that protesting is mainly brought forward as way of expressing resistance rather than demanding implementation.

"Moderator: In an ideal world who should be making decisions on it [SAI]?
Respondent 1: Nobody [...] I'd be out in the street on that one.
Respondent 2: I think governments should shut that one down.
Respondent 3: Yeah.
Moderator: Out in the streets protesting?
Respondent 1: I'd go out, yes. It's just madness, I think, absolute madness.

Moderator: Just clarifying, G2 [Name] you said you'd be out in the street. Do you mean protesting?
Respondent 1: Protesting. Yeah. Oh, yeah, absolutely. Yes."
(Australia Rural)

Supporting civil society associations: public engagement via civil society organizations is considered in almost half of the countries, with a comparable emphasis in Global North and Global South countries. With regard to biogenic CDR, civil society organizations are described as potential implementors, coordinating and supporting afforestation and nature restoration efforts as well as providing spaces for mutual learning and learning-by-doing.

Participants, furthermore, stress the advocacy role of environmental non-governmental and non-profit organizations. They are expected to ensure that decisions are taken with the greater public good in mind and that negative environmental and social impacts of new technologies are minimized. This reflects notions of delegation and indirect public engagement, whereby public engagement with both SRM and CDR methods is channeled through organized and specialized groups.

In some cases, calls for a strong NGO role are tied to lack of political action on climate change due to political cleavages and instrumentalization. As a South African participant suggests regarding SAI governance:

"I think, every country should have like, accredited environmental rights group presenting them, instead of politicians. Because what we've seen, for example, in a country like America, if the governor, or whoever decides global warming does not exist, it doesn't matter what the science says, or whatever it decides, well, I'm the boss, and I decided in my term that global warming is not a problem, then they then pass on all these laws and everything that would undo what the previous one has done, just because this is the state that with the red or the blue." (South Africa urban)

Ambivalence and disinterest: across focus groups in countries with diverse political systems, some participants argue that publics should not have a say and should not play a decisive role when it comes to the governance or implementation of climate-intervention technologies, particularly SAI and space-based geoengineering (SPACE). A particularly strong rejection of public engagement on normative grounds can be observed in discussions across focus groups as the Chinese, German and Turkish ones – albeit for different reasons.

Discussions about whether public engagement with climate-intervention technologies is desirable are tied to the respective national political systems and cultures. For example, the Chinese focus groups do not see themselves in any decision-making role, and describe the role of the public in various ways as promoters and implementors of government decisions.

"If there is anything that needs individual participation, the policy will give us instructions. As said before, there is traffic regulation, which limits car driving for less emission release. There is instruction to tell you which day you can drive according to the last number of you license plate, then I support [follow] it." (China rural)

Some groups' discussions on whether publics should or should not have a say are polarized. This is for example the case for the Turkish focus groups.

Powerlessness: when reflecting on the role of publics, a certain degree of apathy and a sense of powerlessness becomes apparent in some participants' narrations about CDR methods that are perceived

as engineered and technical, and even more so in narrations about SRM approaches. Skepticism and concerns over tokenistic forms of engagement reflect low trust in authorities, the salience of engagement processes, and the intentions of those initiating (top–down) engagements. Numerous such reflections are shared by participants in Australian focus groups, as the following statements regarding the governance of BECCS and DACCS illustrate:

> "I mean, I don't think they're going to care what we say one way or the other, like these town halls and all that sort of stuff. I think they're nice window dressing. And as others have mentioned, they're just going to go ahead and do what they want anyway. Do I think me as an individual gets to have a say that is actually significant? No." (Australia urban)

A participant in a US group summarizes similar frustrations about not being heard by authorities:

> "I think though, locals would like to have a say in this kind of thing. But I think a lot of times when we wanted to have a say in things, even you know, when we like I don't know what to say like protested about it or disagreed with it. You know, they didn't listen to us anyway. So I think that's really the reason why everyone's skipping over oh the local should have a say because I don't think it matters even if we do have a say they're going to do what they want." (USA urban)

## What to engage with

In this section, we report on preferred forms of engagement for different climate-intervention technologies. Despite commonalities, key technological idiosyncrasies mattered. While top–down forms of engagement in issue formation such as information and education were brought forward for most of the approaches (upper left in Fig. 3), citizen-led engagement in implementation (bottom right in Fig. 3) dominated for approaches perceived as low-tech and distributed.

We can infer that conceptions of public agency are intertwined with perceptions of technical simplicity, adaptability and applicability. Participants felt they could most concretely contribute to biogenic CDR methods like afforestation and restoration of vegetation, soil carbon sequestration and biochar. We interpret these narratives as a reflection of participants' efforts of maintaining agency in the context of complex and multi-layered problems. They, furthermore, suggest that more proactive, practical ways of operationalizing might be structurally pregiven by technology features.

In addition to differentiating desirable forms of engagement for varying climate-intervention technologies, publics speak in nuanced terms about the process or stage at which engagement is considered most meaningful. The public preferences for engagement identified here can, thus, also be mapped regarding the timing and object of engagement, reminding of procedural perspectives on participation[68].

Participants – often implicitly – evoked innovation stages, imagining a process from research and formation of the issue via decision-making about rules and standards to implementation. While public engagement on SAI and SPACE was primarily discussed with regard to taking fundamental decisions on whether or not to consider such proposals, reflections on public engagement with biogenic CDR (afforestation, reforestation and restoration of vegetation, soil carbon sequestration and biochar) revolved mostly around the publics' role in implementing and upscaling the approaches. Engineered CDR, particularly DACCS and BECCS and to a lesser degree EW, sat somewhere in-between – and were mostly addressed regarding consultation processes in site-specific and siting-related questions. Showing similarities with discussions about engineered CDR, local or regional consultation was frequently brought forward for MCB.

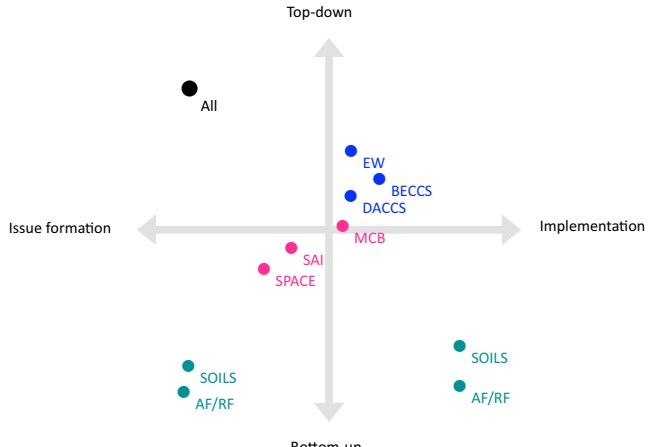

**Fig. 3 | Forms of public engagement emphasized for selected SRM and CDR approaches.** Using a mapping grid adapted from Chilvers et al.[49] the figure displays for the respective climate-intervention technologies which forms of engagement are emphasized across focus groups, whether they are top–down or bottom–up initiated (*y*-axis) and whether they occur in initial issue formation (information; expression of views), decision-making or implementation of climate-intervention technologies (*x*-axis); the mapping shows which types of engagement are most discussed for the respective climate-intervention technologies; categories not mutually exclusive, i.e., different engagement forms can be emphasized for the same carbon dioxide removal (CDR) or solar radiation modification (SRM) approach; mapping is based on authors' interpretation of results in Table 2. AF/RF afforestation, reforestation and restoration, SOILS soil carbon sequestration and biochar, EW enhanced rock weathering, DACCS direct air capture and carbon storage, BECCS bioenergy with carbon capture and storage, MCB marine cloud brightening, SAI stratospheric aerosol injection, SPACE space-based geoengineering.

## Why to engage or not and who to engage

In this section, we report on rationales for and against public engagement and identify corresponding conceptions of publics.

A variety of rationales underpin participants' narrations of why publics should participate in and be engaged with climate-intervention technologies one way or another (Table 3).

Stimulating awareness, learning by doing and fostering agency: corresponding to low intensities of involvement and rather passive constructions of publics, the desire to stimulate awareness is the rationale most consistently brought forward for public engagement across all technology categories, with the highest mentions for SRM in general. This rationale can be found in the majority of Global North and Global South countries. Complementing these rather passive constructions of publics with more active ones and mirroring the comparatively high frequency of "self- and community engagement" for CDR methods, "learning by doing" is a related rationale only mentioned for CDR, and particularly so for biogenic methods. By getting involved for example in the planting of trees and the restoration of vegetation, publics gain practical knowledge and take action.

Giving a voice to those affected, avoiding conflict and ensuring buy-in: participants' calls for consultative approaches are underpinned by a variety of rationales. Participants differentiate between giving publics the possibility to express their opinions on the approaches at a general level, and about hearing those who would be directly affected by a specific project or implementation plan. The former is particularly tangible in calls for consultations on SRM approaches for which – given the perceived high stakes and global impacts – participants argue that publics should be able to express their opinions, while the latter is more pronounced for CDR methods.

Reflecting the latter rationale, one participant from Sweden argues that the governance of EW requires consultations in order to

**Table 3 | Summary of rationales for and against public engagement mentioned across focus groups and technologies**

| Rationale | | Technological focus | Frequency across countries |
|---|---|---|---|
| For | To raise awareness | SRM in general | High |
| | To foster learning by doing and stimulate agency | CDR in general, AF/RF, SOILS, no mentions for SRM | Medium |
| | To ensure affected parties are heard | SOILS, DACCS, EW, SAI | Low |
| | To avoid conflict, engage with opposition & secure support | DACCS, BECCS, SAI | Medium |
| | To keep big business in line | SRM in general | Low |
| | To pressure government to act | CDR in general, SAI | Low |
| | To gain knowledge | CDR in general, AF/RF, SRM in general | Low |
| Against | Too complex & too technical, incl. risk of manipulation | SRM in general, SAI, SPACE | High |
| | Other priorities of publics | AF/RF | Low |
| | Trust and support government, often linked to complexity | DACCS, SAI, SPACE | Medium |
| | Makes decision processes inefficient, slow or impossible | BECCS, but emphasis not pronounced | Low |

Low frequency = mentioned in 7 countries or less; medium frequency = mentioned in 8–14 countries; high frequency = mentioned in 15 countries or more "SRM in general" and "CDR in general" are used where participants refer collectively to SRM or CDR, without specifying which particular method or technology they speak to.

SRM solar radiation modification, CDR carbon dioxide removal, AF/RF afforestation, reforestation and restoration, SOILS soil carbon sequestration and biochar, EW enhanced rock weathering, DACCS direct air capture and carbon storage, BECCS bioenergy with carbon capture and storage, MCB marine cloud brightening, SAI stratospheric aerosol injection, SPACE space-based geoengineering.

give a voice to affected communities, particularly indigenous communities:

> "Exactly, that the community that is being affected, that it is important for them to have a voice. I'm thinking this in relation to 'biting into the sour apple', that it is at the expense of a lot of people, and that it might be due to some other cost. If you are to take, the mining example, for example, the Sami land and reindeer fencing and all of- how that is affected because the mine is expanding. So I think you need a holistic perspective and that you need a chance to affect as a local community or the population of a local society." (Sweden rural)

Such rationales are considered all the more important given that the large-scale implementation of CDR methods might interfere with attachments to place and heritage in some communities. One Polish participant explains:

> "But if there are no such broad consultations, first of all honestly, then nothing will come of it. (…) I'm sure there will be hundreds, millions of people that would need convincing, because, as we said, there are dozens, hundreds of farmers who will have to be paid for land for planting trees, or people who own wasteland but are emotionally attached to the land. I mean, this is an issue mentioned earlier, people in Poland are tremendously attached to the land they own, their family heirlooms, so 'even if it does not produce crops and does not bring any income, it is my patrimony, and I will not have it covered with stones'." (Poland rural)

As with the idea of giving a voice to those affected, consultative approaches are also underpinned by the rationale that involvement would help avoid conflict and allow to engage with opposition early on. This is most mentioned for public engagement with engineered CDR methods (e.g., BECCS and DACCS), reflecting discussions about perceived public acceptance issues and NIMBY-ism.

> "I always sort of think it is a matter of whether you include the entire population or the affected population. […] With DACCS,

the argument was if this big plant is placed in my neighborhood, I would like to be involved in the decision making. I do think that the affected populations should be included, just to make it functional long-term. It will not be tolerated long-term, if that is just decided behind the backs of the people, who it affects." (Germany rural)

> "I just wonder, I'm not saying that because of where I live because I live in London but, for example, - and I know the planet is more important than our back gardens - but I wonder if there is a place with fabulous views of fields and then a factory is going to be built there, I might like to have some kind of say in that." (UK urban)

Gaining local and experience-based knowledge and inform assessments: primarily regarding biogenic CDR methods and MCB, participants speak about gaining local or experience-based knowledge as a motivation for engagement processes, arguing that this would ultimately help with improving assessments and decisions about how and where to implement climate-intervention technologies. In this rationale, participants mostly talk about specialized publics rather than the public in general – defined by the knowledge that they hold about relevant practices or specific places, as the following example about local knowledge relevant for assessing MCB on the Great Barrier Reef illustrates:

> "I think that there should be input from the local area, but maybe not down just to the general people in the area, but more taking each individual, perhaps council area or specific thing as a barrier reef, and looking at that, so the people who are in the position that they're looking after the Barrier Reef, they should have some input too." (Australia rural)

Similarly, regarding afforestation and restoration of vegetation as well as soil carbon approaches, experienced-based knowledge of rural communities and indigenous groups is acknowledged and should be elicited in engagement processes, as exemplified by the following extracts from an Indian and a Norwegian focus group:

"From rural communities actually, I think they will be more helpful, and they will contribute more because they have always been rooted on the earth. So, they know what crops to grow and what fertilization should happen. They work at the ground level so they will be able to guide the scientists or guide anybody from the grass root level I feel." (India urban)

"Then I also think it's important in many parts of the country to use those who know the earth, who know the vegetation. Like indigenous people, farmers, who have carried on traditions for hundreds of years and can tell you how it was here a long time ago. There's a lot of knowledge there." (Norway rural)

Further rationales underpinning arguments for public engagement include the idea that this would allow publics – through civic organizations or pressuring governments – to maintain (regulatory) oversight over industry and corporate activity.

Participants also identify various rationales why publics should not engage in the governance of climate-intervention technologies – at least not in any decision-making roles (Table 3).

Complexity and technical nature of the approaches: the complexity and technicality of climate-intervention approaches, and the lack of publics to sufficiently comprehend their implications, is by far the most discussed rationale underpinning caution on (direct) public involvement in governance. This rationale is mentioned with regard to technological CDR methods, but is emphasized most strongly with regard to SRM approaches. For SRM other intersecting rationales brought forward against an active role for publics include the perception that publics are easily manipulated, as well as the exceedingly high stakes of decision-making. This rationale is frequently mentioned across groups, with a particularly high salience in Germany, followed by groups in Australia, the US and Switzerland. Participants in European focus groups draw comparisons to the Brexit referendum in the UK. Similarly, participants make analogies to the COVID-19 pandemic to argue that uninformed publics may make irrational decisions. In both cases, participants fear that politicization and populism render public engagement with SRM risky, as the following extract from an Austrian focus group suggests:

"Unfortunately, I have to use Covid vaccines as a negative example. There was also a huge scientific debate, and so many people who weren't vaccinated because certain powers in our democratic countries know that they can play with fears and move the discourse away from science and towards emotions to make a profit. It shows us that beforehand it requires… I must say I'm sceptical about a peoples' vote because you see what can happen. Take Brexit, where people are misled to make idiotic decisions. I think it requires a low-threshold, broad educational campaign ahead of time. To me, this is the only way to get people on this side and to take the power away from right-wing populists." (Austria urban)

This rationale is coupled with expressions of high trust in experts and governments to take well-informed and well-intentioned decisions. Since in this rationale publics do not have anything meaningful to contribute, decisions about complex issues such as CDR and even more so SRM technologies are arguably better handled by experts with specialized and certified knowledge, as suggested for the case of SPACE by this participant in a Spanish focus group:

"I think the decision should be made by experts; we shouldn't make the decision. We all have different backgrounds, and it should be scientists or engineers who make the decision.

They should inform us, but it should be up to specialists to make the decision." (Spain rural)

In these narrations, it becomes clear that much of the construction of publics takes place in relation to the construction of other actors, most notably "the experts". Focus groups participants tend to express high trust in scientific experts. Participants' reflections on the (limited) role of publics are embedded in calls for a strong role of experts – not only as providers of evidence or holders of knowledge, but also as decision-makers.

At the same time some participants question the integrity of experts or emphasize that their objectivity and impartiality need to be ensured, suggesting that they might have experienced otherwise in the past, or at least that some form of democratic legitimacy needs to be ensured. Reflecting on the role of experts in the governance of SAI, one participant argues:

"And this needs to be scientifically based, but you can't let the scientists run free, there needs to be an organization, like the UN, above them that runs the show. But it needs to be based on science and research." (Norway rural)

In certain – mostly European – focus groups, reflections about expertise and the perceived lack thereof in the general public give rise to fundamental discussions about democratic legitimacy, and how the right to vote must not be undermined by criteria of any sorts, including the degree of knowledge and education. The following debate between two German participants illustrates this tension:

"Respondent 1: It also matters if it is about information or actually about decision making. In principle, I am very democratic, but you would have to somehow ensure that the knowledge is actually available. I cannot expect a population to make an informed decision without this knowledge first being acquired. You could possibly first allow experts to research it, to find out what all the risks are… however, I would not want people to vote, who do not believe climate change exists. Because unfortunately, they do not have the information and knowledge about how science works. […]. I do not have a solution, but I think you would have to ensure that an informed decision is made.

Respondent 2: Yes, but you cannot say, they cannot be part of the decision because I think they are dumb or they are conspiracy theorists or whatever. That does not work in our democratic system. […] I agree with your thoughts and your reservations, but that is not practicable. We cannot say, you cannot vote because you do not know, you are part of the wrong party, you are a conspiracy theorist or have no idea or have never looked into that. We just cannot exclude certain groups. […] It is a problem of democracy." (Germany rural)

In other focus groups – particularly in non-democratic systems such as China and Saudi Arabia – rationales against public involvement are intertwined with expressions of high trust in governmental actors to take well-informed decisions in the best interest of citizens, as this Saudi Arabian participant explains:

"The government knows what is best for the people […] and they work for the general benefit of the people and know what they should do when it comes to vegetation and the health of it and the health of people." (Saudi Arabia urban)

Inefficient, difficult or impossible decision and implementation processes: participants across focus groups – though most strongly in European, democratic, countries – express concerns about the inefficiency or even impossibility of decision-making with strong public involvement. Procedural concerns play a considerable role in publics' considerations. Direct-democratic decision-making is considered implausible and too complex to implement, particularly at a global level – which is seen as the main governance level for approaches such as SAI. Adjoining rationales include the lack of feasibility from practical and organizational points of view and the unlikelihood of finding an actionable consensus. Examples of perceived failure to find international common ground on climate action serve as a case in point – between states or of prospective processes that would involve citizens worldwide. The following statement exemplifies this skepticism toward implementing global public engagement in decision-making about SRM:

"Our role is to keep our fingers crossed that this will be successful, because this is being done on such a large scale, it's hard for there to be unanimity of all the citizens of the world here. Everyone agrees and everyone raises their hands, yes? It can't be like that!" (Poland rural)

Inertia, inefficiency, and negative impact on the lengths of decision-making with public participation are also discussed regarding CDR methods, particularly those perceived as more technical and engineered. Participants associate them with the local level, where opposition and resistance might manifest strongly.

"[I]f I involve the general public, it will become like chewing gum [a tedious, unworkable process]. It will get dragged out more and more and more before you finally have a result. If I want to make progress, I need shorter pathways, quicker decisions and to not blow it up within the broader population via voting and whatever else. That will not result in anything, at the end of the day". (Germany rural)

Parallels are drawn to slow implementation processes in the energy transition and analogies to photovoltaics and wind turbines are made, one Norwegian participant refers to the "windmill syndrome" in this context. A closer look at the discussion dynamics in the Swiss focus groups, for example, point to the need for understanding perspectives on public engagement in the context of local experiences with public participation, political cultures and systems:

"In Switzerland, the problem is that the population has got quite a lot to say about it, and as soon as it is handled by companies, resistance will immediately arise and this will drag on for years, if not decades. It is incomprehensible to me why wind energy is being fought on a massive scale in our country. I would have no trouble having a wind turbine 500 meters away from my house. In Switzerland that is not feasible, they are being fought so much just because someone feels that they don't want to see it from their balcony. It will be exactly the same as soon as companies start implementing such [CDR] projects. Resistance forms immediately and the projects are being fought". (Switzerland rural)

Less mentioned rationales against public engagement include arguments that publics have other priorities to focus their efforts on, or that they do not see any substantial benefit from engaging.

We suggest that different rationales for and against public engagement are accompanied by different conceptions of who the public is. Resembling what Michael[65] called "publics in general" vs. "publics in particular", participants differentiated among types of publics (Fig. 4), delineated by knowledge or stakes and degrees of affectedness. These findings suggest that engagement processes need to be tailored, depending on whether the general public or specific host communities are targeted – supporting similar conclusions drawn for the case of DACCS in the US[82]. Likewise, rationales underpinning the rejection or disinterest in public engagement mirror views of how the public is (e.g., ignorant, depoliticized or resisting) or how institutions are (e.g., not trustworthy). They, thus, offer insights into deeper roots of non-engagement and political apathy.

## Discussion
In the following, we discuss the main contributions of this analysis to the emerging literature on public engagement with CDR and/or SRM, as well as to participation scholarship more generally. Previous research identified the need for public engagement with selected CDR[33,36,37] and/or SRM approaches[20,21] and found a general interest of publics in information and engagement campaigns[47]. Our results show that a variety of engagement options can accompany the deliberative exercises that social scientists working on climate-intervention have been focusing on, and contribute to an "expanded role for deliberative thinking"[38].

The most discussed forms of public engagement across technologies and countries were as follows: receiving more information, self- and community engagement and consultations. In participants'

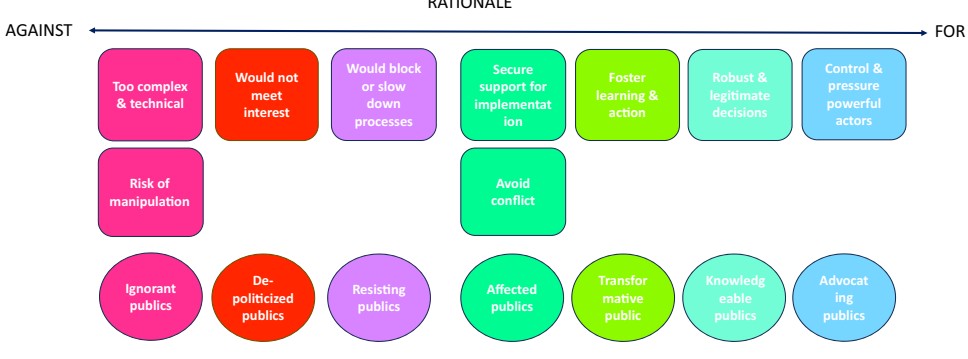

**Fig. 4 | Summary of main rationales for and against public engagement raised in focus groups in 22 countries and corresponding conceptions of the public.** We combined some of the inductively developed categories shown in Table 3; conceptions of the public are our interpretation of the underlying, sometimes tacit conception present in focus group participants' speech; red and pink tones signal rationales against engagement, blue and green tones signal rationales for engagement.

narration these appear not as either or propositions, but rather as overlapping or sequenced engagements, where more intense or active forms of engagement can build on less intense and more passive initial activity. Mirroring well-documented low levels of familiarity with climate-intervention technologies[40,83–85], more knowledge and information is demanded particularly for the SRM approaches, but also for CDR. Knowledge and information are identified as an important precondition of any form of meaningful engagement with climate-intervention technologies, suggesting cascading intensities of involvement.

While the knowledge deficit model[45] as well as various levels of participation[21,69] are raised by members of the public themselves, they envision further ways of engaging not traditionally captured in such participation theories. For example, references to practical ways of engaging with CDR methods and climate action point to the importance of considering wider "ecologies of participation" for understanding the diverse ways in which publics engage with emerging technologies[76], suggesting parallels to similar mappings for the case of energy systems[86,87]. Additional reflections on individual efforts not directly related to any of the climate-intervention technologies, such as changes in consumption and mobility practices, echo the responsibilization of individuals and an emphasis on individual behavior changes as drivers of transformation[88], while neglecting the interplay with structural and systemic dynamics of (un)sustainable practices[89].

In the context of typologies of participation[50], our results show that engagement with climate-intervention technologies is considered desirable for instrumental (e.g., increasing acceptance, reducing opposition), normative (e.g., those affected should have a right to express their opinions) and substantial reasons (e.g., specific local knowledge can improve decisions about implementation and reduce potential trade-offs). At the same time, some participants are sceptical about public engagement. Discussions surrounding particularly intense forms of engagement and direct decision-making were polarized. Reminiscent of governance-related concerns brought forward by proponents of a non-use agreement for solar geoengineering[90], many participants, particularly in European countries, were concerned that finding agreement at a global scale for governing SRM would be highly unlikely, even more so when involving publics worldwide.

As the climate crisis accelerates, urgent political action and technological promises and advancements render the development of responsive and inclusive interfaces between science, society and policy ever more important[49]. In the context of novel and emerging climate-intervention technologies, developing such interfaces also means moving from abstract calls for public engagement to the development of situated engagement practices that are sensitive to local political cultures and socio-technical environments, while attending to the global dimensions and interconnectedness of the issues at stake. Echoing Perlaviciute's[51] conclusion for climate policies, our results overall suggest that a better understanding of public preferences for participation is critical for developing governance frameworks for emerging climate-intervention technologies that are perceived as just, ethical and effective.

In conclusion, we outline conditions for meaningful public engagement as a cornerstone of inclusive and sustainable governance of these approaches. Several key considerations for meaningful public engagement emerge from our analysis, particularly relevant for engagement processes orchestrated by public authorities:

**Recognizing that participation exists across an ecology of subjects, objects, and models:** some participants ranged from hostile to ambivalent about engaging with climate interventions, whereas others wanted to be fully informed, while some envisioned active roles (e.g., direct action). Some prescribed a valued role for civic debate; others prescribed a narrower role, whereby predominantly experts should be engaged. The issues and concerns at stake vary in intensity and severity as well, and touch on themes including power relations,

democracy, scaling, and perceived dangers in the form of externalities and risks. The proposed or even preferred modes of participation also vary greatly, ranging from passing out leaflets and volunteering, to townhall debates, being a respondent in surveys and opinion polls, voting, campaigning, and demonstrating, or even referenda and formal petitions. The particulars of the ecologies of participation for climate-intervention technologies thereby transcend the divides between institution vs. citizen-led efforts and capture a diversity of practices and settings by which publics intend to engage, or not, on such options.

**Accounting for national contexts and modes of civic engagement:** in some cases, reflections on the desirability of particular forms of engagement reveal experiences with (local) participation cultures and are deeply rooted in participants' embedding in national political systems. Some emphasized expected, even formalized modes of civic engagement – for example, Indonesian participants ubiquitously noted the need for "socialization" (a well-used term that forms part of a national perspective on education and engagement), the Swiss referred to experiences with direct democracy, and Global North Anglophone groups all brought up "townhalls" as a venue for deliberative municipal politics. Contestation over the role of publics in decision-making also exemplifies this point. For example, Global North perspectives on public agency were often offset by reflections over NIMBYism. Others voiced examples of political tribalism or local parochialism – e.g., Swiss "cantonialism" or "red vs. blue" dynamics in the US. Meanwhile, similar skepticism over the need for intense public engagement from participants in China and Saudi Arabia were tied to high trust in the capacity of their governments to take decisions in the public interest, and mirrored the limited role of publics in non-democratic political systems. Conversely, political corruption was voiced across a variety of national systems as a context for skepticism regarding the value of public consultation processes, ranging from Poland and Italy to the Dominican Republic, Brazil, and South Africa. We caution that national specificities were due to our focus on cross-country participatory modes not strongly analyzed here. Our observations, however, suggest that they require further attention.

**Tailoring engagement practices to specific technologies:** public preferences for engagement vary depending on the climate-intervention approaches in question. For more distributed biogenic CDR methods such as reforestation and nature restoration, formats that encourage and facilitate active and practical engagement in local implementation and deployment efforts should be expanded. For more centralized, engineered CDR methods such as DACCS, BECCS and to a lesser extent EW as well as MCB, community consultation and public involvement in government-industry decision-making on infrastructure siting, transportation, and carbon storage, as well as the design and risk management of initial demonstration projects, are central. For SRM approaches – particular those with planetary impacts – widespread and direct public participation in decision-making about whether these approaches should be considered are strongly called for by some, and vehemently opposed by others. Under conditions of such uncertainty and polarization, there is a need to create global spaces for deliberation and debate that are – for now – free from pressures of taking policy decisions or finding consensus.

**Developing power-sensitive practices:** a gap between what publics consider desirable and what they consider realistic regarding their participation in developing governance structures and/or implementation plans for climate-intervention technologies became apparent in this study. This gap is tangible in accounts of disillusion and feelings of powerlessness as well as narrations of not being consulted or of not being heard when powerful vested interests are at play. Developing inclusive public engagement with emerging climate-intervention technologies, thus, necessitates a critical interrogation of past and present forms exclusion and power dynamics in decision-making processes and in responsible innovation more generally[48].

Adopting power-sensitive practices also means acknowledging and tackling unequal opportunities to engage[91] as well as recognizing that some publics might not want to engage actively with such emerging technologies and prefer to delegate tackling questions surrounding them to either elected representatives or to those perceived as experts, hence expressing preferences for clear divisions of roles and responsibilities between different stakeholders[92].

**Co-producing knowledge and accounting for prior experiences:** participants across countries demand more knowledge about SRM and CDR approaches and consider a better knowledge base as a precondition for any other, more intense, form of public engagement. While the provision of knowledge and information appears as a key ingredient of meaningful engagement, information and engagement campaigns should not be designed in a "deficit logic"[45] and publics should not be considered as "empty vessels" that, once filled with information and knowledge, will "think as experts do"[63]. Rather the multiple analogies and references to technologies perceived as similar and to past political (engagement) processes deemed comparable suggest that co-production and communication of knowledge that accounts for publics' prior knowledge and experiences can contribute to more effective engagement[93]. Such a move toward a co-production approach is underscored by our results showing that participants describe "publics in particular"[65] as holders of valuable knowledge, particularly salient for biogenic CDR methods and as brokers translating knowledge about "engineered" climate-intervention approaches to wider publics.

**Establishing trust and procedural legitimacy:** trust appears as a central theme in participants' reflections on public engagement and the governance of climate-intervention technologies more generally. Fostering trust in institutions through transparent communication of side-effects and uncertainties, involving trusted experts as well as early clarification of how outcomes of the engagement process will feed into decision or implementation processes emerge as important elements in a "chain of trust"[94] that can enhance perceived procedural legitimacy. At the level of projects setting up trustworthy processes also requires dedicating resources to community involvement, as for example suggested for DACCS[82].

**Engaging constructively with dissent, tensions and value disagreement:** participants' frequent reference to challenges and even the impossibility of finding common ground on (the governance of) climate-intervention technologies, particularly at a global scale, point to the need for engagement processes that value debate and negotiation over consensus and unanimity. Engaging with the diversity of perspectives on climate-intervention technologies also means being attentive to bottom-up and claimed spaces of participation and engagement, for example when controversies lead to protests. Following this reasoning controversies, conflicts and protests are – unlike in a public acceptance perspective – not problems that need to be avoided but can rather be seen as instances of social learning[49] and as informal, bottom-up assessments of technologies[95] that might complement more formalized, scientific assessments.

## Methods
### Data collection and analysis
This paper is based on the qualitative analysis of 44 focus groups conducted in 22 countries (with one focus group in each country in a rural and one in an urban environment), involving in total 323 participants with diverse socio-demographic backgrounds (Fig. 1). Recruitment for and implementation of the focus groups were done in collaboration with Norstat, a European-based data collection company.

The preparatory reading material for participants and the discussion guide included a variety of climate-intervention technologies. We chose two biogenic sets of carbon removal methods: (1) afforestation/reforestation and restoration of vegetation, as a proxy for

management of terrestrial and marine ecosystems, including blue carbon, (2) soil carbon sequestration, as a proxy for agricultural management practices, including biochar. We then chose two distinct chemical carbon removal methods: (3) direct air capture and carbon storage (DACCS) and (4) enhanced weathering. Finally, we opted for (5) bioenergy carbon capture and storage (BECCS), a hybrid system that combines a bioenergy input with a technical storage component. In addition, three SRM approaches were included: stratospheric aerosol injection (SAI), marine cloud brightening (MCB), space-based geoengineering (SPACE). We acknowledge debates on the conditions under which these two broad suits of approaches should be separately (representing different socio-technical characteristics and governance demands) or comparatively assessed (for synergies and trade-offs in the context of wider climate action). With the expectation that they offer contrasting and overlapping cases for publics' self-conceptions of their roles and agency, we opted for including both in this article.

As part of a wider discussion on relevant actors and actions in the governance of these climate-intervention technologies, one guiding question was specifically designed to stimulate reflection on the role of the public: "How would you want yourself, and the wider public, to be involved in making decisions on these approaches?"

All focus groups were recorded, transcribed verbatim and translated to English. The empirical data were managed, coded, and analyzed with the software "MAXQDA". We conducted thematic analysis[96] of the focus transcripts, combining deductive and inductive steps. While the above-outlined conceptions of participation and engagement guided the development of the main themes, more fine-grained and emerging themes were inductively developed (Supplementary Figs. 1–4). During the analysis, these participation and engagement-focused categories were crossed with the CDR and SRM approaches discussed in the focus groups (Table 4). The first and second author were responsible for coding the data. Investigator triangulation relied on a "negotiated agreement" approach to establish inter-coder agreement[97].

In reporting the results, we focus on recurring themes that are addressed across various focus groups. Where possible we emphasize context specificities or variances in interpretation of themes across groups. While counts of country mentions offer some indication of the relevance and salience of themes, group dynamics may simply have led to a focus on certain themes and a neglect of others. References to country mentions and counts thus need to be interpreted with caution. Given the multi-technology, multi-country approach of this study in-depth discussions of individual technologies and countries cannot be provided in this article.

A detailed description of research design is provided in the Supplementary Information (Supplementary Methods).

### Institutional review board approval
All components of the research were granted ethical approval by relevant authorities at Aarhus University (#2021-13).

### Ethical review statement
Full and informed consent was given by all participants before the beginning of the study, along with all participants being notified about the fact that their data would be handled in a fully anonymous manner and in complete accordance with the General Data Protection Regulation and any other pertinent data-security regulations, that any data would be analyzed in an aggregate fashion and would not be personally identifiable in any way, and that they had the right to withdraw their participation at any time.

### Inclusion & ethics statement
All components of the research were granted ethical approval by relevant authorities at Aarhus University (#2021-13). Full and informed consent was given by all participants before the beginning of the study,

**Table 4 | Analytical framework guiding the development of dimensions of public engagement used for analyzing focus group transcripts**

| Analytical question | Analytical category | Examples of inductive sub-categories | Related engagement categories in literature | |
|---|---|---|---|---|
| WHY: what motivations underpin publics' reflection on whether and which forms of engagement are considered desirable? Why is public engagement considered (not) desirable? | Rationales | To avoid conflict and engage with opposition; to ensure affected parties are heard | Typology on instrumental, substantive, normative rationales[52] and various adaptations, e.g.[98] | WHAT: To which climate-intervention technologies and to which processes do reflections about public engagement relate? (Objects of participation[76]) |
| HOW: in which ways should (or does) public engagement occur? | Forms and intensity | Information, consultation, decision-making | Ladder of participation[66] and its various reinterpretations, e.g.,[67–69] institution-led vs. citizen-led practices[49] | |
| WHO: who is seen as the relevant public, how is "the" public characterized and delimited? | Publics | Ignorant publics, affected publics, resisting publics | "Publics in general" and "publics in particular"[65] "obligatory publics"[64] | |

along with all participants being notified about the fact that their data would be handled in a fully anonymous manner and in complete accordance with the General Data Protection Regulation and any other pertinent data-security regulations, that any data would be analyzed in an aggregate fashion and would not be personally identifiable in any way, and that they had the right to withdraw their participation at any time. The research has been broadly undertaken with the aim of better understanding public perceptions of carbon removal and SRM approaches, including in the Global South and by means of more qualitative methods that can better elucidate the variability and importance of the local context. At this stage, no local researchers have been included. As noted in the *Ethical Review Statement*, this research has been approved by the ethics review committee; in addition, the specific roles and responsibilities of those in the author team were discussed prior to the research. Insofar as possible, we have also strived to take into account local and regional research in the citations.

### Reporting summary

Further information on research design is available in the Nature Portfolio Reporting Summary linked to this article.

## Data availability

The dataset generated and analyzed during the current study is not publicly available at present, to allow for further analysis and publication of findings over the course of the project. Access to the raw data can be made available from the corresponding author on reasonable request. The totality of the dataset will be made publicly available in full following the conclusion of the GENIE project.

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

## Acknowledgements

The authors acknowledge funding from the European Union's Horizon 2020 research and innovation programme under the European Research Council (ERC) Grant Agreement No. 951542-GENIE-ERC-2020-SyG, "GeoEngineering and NegatIve Emissions pathways in Europe" (GENIE) as well as under Grant Agreement No. 101081521, "Bridging current knowledge gaps to enable the UPTAKE of carbon dioxide removal methods" (UPTAKE). The content of this deliverable does not reflect the official opinion of the European Union. Responsibility for the information and views expressed herein lies entirely with the author(s). The authors acknowledge the numerous colleagues who helped translate key terms into different languages for the information materials.

## Author contributions

L.F., S.L., C.M.B and B.K.S. designed the study. L.F. and S.L. undertook data analysis and synthesis, with input from C.M.B. L.F. wrote the manuscript, with content and reference inputs from S.L. and C.M.B. L.F., S.L., C.M.B. and B.K.S. edited the manuscript to completion.

## Competing interests

The authors declare no competing interests.
