## [Peer Review File · Nature Communications]

Public engagement for inclusive and sustainable governance of climate interventionsReviewers' Comments:

Reviewer #2:

Remarks to the Author:

The article examines public preferences and rationales for engagement practices in carbon dioxide removal and solar radiation management. The question addressed is What do diverse publics think about their own role when it comes to decision-making about CDR and SRM technologies? The paper is based on the qualitative analysis of 44 focus groups across 22 countries. It finds various forms of public engagement and rationales for or against participation around CDR and SRM. The article concludes that engagement practices (1) should be adopted to specific methods, (2) reflect on power dynamics, (3) account for publics' previous knowledge and experiences, (4) foster trust in processes and institutions, and (5) value debate and negotiation. The paper's results and conclusions seem isolated and are not particularly novel yet. However, the paper shows promise due to its broad geographical coverage and its contribution to the expansion of the idea of participation applied to CDR and SRM. I recommend the article be accepted pending major revisions, which I detail below:

General

-The paper could be strengthened by a more careful reflection on which dimensions are introduced in each section. In the results, you report on the forms and rationales of public engagement, and how the public is delimited (how, why, who) for different places and technologies (where and what). In the discussion, the "when" is introduced and the "where" is omitted. In the conclusion, considerations for meaningful public engagement are introduced. A more careful selection of which dimensions are present in each section, which of their interactions are highlighted and how the argument you construct leads to the introduction of new dimensions would help readers to follow your train of thought.

-You use the terms "method" (e.g., line 75), "technology" (e.g., lines 27, 86, 88, "approach" (e.g., lines 17, 116, 179) interchangeably. Clarify which term you use when you refer to both SRM and CDR, which one when you refer to CDR and which one when you talk about specific CDR methods. I suggest using the term "method" for CDR, following the terminology of the IPCC AR6.

Introduction

Line 37. Typologies that divide CDR methods into "ecosystems-based" and "engineered" are typically problematic (See e.g., Osaka et al, 2021). It would be more accurate to list the methods without trying to force them into a typology.

39. You include the definition of SRM (offsetting global warming by reflecting incoming sunlight). Include the definition of carbon dioxide removal as well.

40. I recommend nuancing this point. Discussions on the scale and role of the publics in approaches of, for example AR, and the different composite parts of BECCS, are not new. See, for example, <https://doi.org/10.1002/wcc.671>

78. I suggest mentioning here or elsewhere that certain forms of involvement are given structurally by the methods. For instance, soil carbon enhancement can be done by farmers and other individuals, while industrial capture or underground storage components cannot.

84. It would probably make sense to describe what you mean by "role" before introducing it in the question. This might clarify how it relates to the "who", "how", and "why" dimensions you explore in the paper.

84. In the results and methods you indicate that the paper explores not only how people see their own role but also that of their fellow citizens. Make sure you are consistent between your RQ and the results your report.

Conceptual background

104. This sentence is very long. Consider splitting it into two or more sentences.

107 Mentioning DACCS here sounds odd, because concerns over geological storage and siting do not only concern this method. I suggest referring to geological storage only or to both DACCS and BECCS.

136 This table is very useful to guide the reader.

153 To say that "In the field of climate and environment-related decision-making, empirical evidence for such desired, substantive, outcomes of participatory approaches is limited» is quite a bold statement. This will probably depend on how you define «outcome», but I would argue that much of the work of Elinor Ostrom, for example, demonstrates the opposite. I invite you to nuance this statement or include references that support it.

Research Methods

262. Clarify how the recruitment of participants was done, and whether they received any compensation.

277. I am not sure that these links to other methods is necessary. The link to DACCS is particularly confusing.

307. It is not clear to me why the analytical category «forms and formats» in table 2 has a different name from the respective approach «on degree or intensity» of table 1 and the section «forms and intensities» in the Results section. Clarify the difference or use the same terminology throughout the text.

307 Add punctuation marks to the citations. Add year to Chilvers et al.

Results

-I suggest that you report on the interrelationships between the dimensions of analysis already in this section, rather than (only) keeping them in isolation. For example, by presenting a table where the forms, rationales and constructions of the public are presented together, by moving some of the graphs you present in the discussion to this section, or by exemplifying some of the "ecologies" you observe around different methods in specific regions.

-Given that you have a unique dataset from a diversity of countries, it would be valuable if you could summarize the differences you observed between them. This could provide a more nuanced picture of contexts of participation than generic typologies.

-In my opinion, there are too many quotes in the text and some of them are too long for the points you want to exemplify. I invite you to select only the quotes you find most insightful and, when possible, keep them rather short.

374 You write that "Narratives about self-engagement reflect efforts of assuming agency and taking

responsibility". Is this part of your interpretation?

392 In line with the other approaches, clarify how you defined consultation approaches.

419. Replace DAC by DACCS. DAC is not CDR.

485. I find this dialogue confusing. Perhaps it is possible to omit the parts that do not add to the argument?

567 In some cases, it is not clear to me what you mean by "in general" and "mentioned for all" in the table. For example: "Afforestation and restoration, Soil carbon sequestration and biochar, CDR in general; mentioned for all".

570. This subtitle should probably refer to both the rationales and the "who".

570 The analytical lens you use to build the rationales is not clear to me. It seems as if they were rather organized according to the different types of participation (with more or less empowerment) than according to, for instance, the rationales of Fiorino, 1989.

790. Some of the contents of this table seem to be absent in the text or lack an explicit connection (e.g., the rationale "other priorities of publics", the constructions of transformative and resisting publics). Make sure to maintain consistency between the interpretations in the text and the categories in the table. Speak in the first person when describing your interpretations. So far, the authors seem to be absent in the text.

Discussion

-The diversity of countries is a key feature of your paper but is largely absent from this section. It would be useful, for example, to discuss how the culture and history of countries, in general terms, determine the forms of participation mentioned by participants.

-Although you describe your results as contributing to the "expanded role of deliberative thinking" and highlight the importance of "ecosystems of participation", the relationship of your results to these concepts would benefit from a richer discussion.

803. Clarify in more detail how your results contribute to the "expanded role of deliberative thinking", in comparison to "conventional" ideas of public engagement and existing systemic mappings of participation, like the ones of Chilvers et al, 2021 and Chilvers et al, 2023. How does this represent a new way of thinking about participation for decision makers, citizens and/or social scientists working on CDR/SRM?

812. The dynamics of participation over time is an interesting point to mention! You can return to the work of Stauffacher et al. (2008) here. Much of the information here is new and could therefore be included in the results.

825. Some examples of the ways of participation mentioned by the publics that are not traditionally captured in participation theories would be helpful here.

826. Do you refer to "ecologies" of participation?

830. The role of the citizens, how they see themselves interacting with different types of technologies and why could already be brought up in the results.

839 As previously said, I suggest moving these figures and text to the results, if the “when” is part of the role of the publics you are exploring.

839 On the Figure II, B: 1) Why is the axis called self-engagement? Is it restricted to the category of “self- and community engagement”? 2) It is not clear to me why you have two dots for soils and AF/RF (on issue formation and implementation) and only one dot for the other methods. Does the dot for “All” exclude AF/RF? 3) The graph suggests that for methods like BECCS, DACCS and EW the most discussed type of engagement is non-engagement. An expansion of this on the text would be relevant. 4) Clarify how the intensity of discussion is measured for this figure.

882 As mentioned before, this might also be given structurally by the methods.

882 Do you observe in your material that public agency and technical simplicity are coupled or is this your interpretation?

898 I encourage you to elaborate on the participants' scepticism about public engagement. What is the reason for this discrepancy among participants? What does this tell us about the design of participatory processes for CDR / SRM?

Conclusions

-The conclusions do not sound very novel yet. However, the material offers such rich sources, that there must be much more nuanced and specific that can be said. Some of these points (e.g., on power and agency) and how they shape dynamics of participation across methods could be taken up more strongly in the discussion.

Discuss in light of what is new here, how it builds on existing literature, how it expands the concept of ecologies of participation or public engagement and links it to CDR and SRM.

Appendix

-If possible, include in the preparatory reading material provided to participants.

Reviewer #3:

Remarks to the Author:

Overview:

This paper provided decent insights into participation of publics across 22 countries in both the Global North and the Global South in considering governance of climate intervention technologies. The introduction explores the literature to help contextualise interpretations of what ‘participation’ looks like. The rationale of the paper is sound and the framework developed is clear but it is hard to see how this could be replicated. The methods left a lot of questions for the reader even taking into consideration the supplementary information. Participants were given a week to read information given to them on a range of different techniques across both SRM/CDR but it was not clear if they had engaged with the material or done their own research especially considering the range of techniques discussed over the two hour session.

Introduction:

Overall a thoughtful overview of participation of publics in the emerging technologies space.

Methods:

I acknowledge there are 22 groups in the sample but two hours seems like an incredibly short time for

a workshop discussing complex and novel information for participants.

The authors describe the content of the information in the appendix. Could they include the actual material content rather than the description provided of what it contained?

Is there any evidence that describing what the authors have termed 'biogenic' approaches as those 'that change how we use nature' changes how people will respond technologies by not using 'nature-based'? In my mind the updated phrasing seems similar to a 'messing with nature' framing. In addition the word nature itself can mean many different things to different people and cultures.

Were the translated (and original) materials piloted before they were used in the actual workshops?

(Based on the previous comment that terminology and phrases can be interpreted differently between countries and cultures so hard to know how comparable different strategies may be).

In the description of the material it says the pros and cons were abbreviated as much as possible to limit framing, it is not clear what this information was or how it was abbreviated?

Results:

Line 485 – 497 – how does the reader distinguish between respondents and/or moderators in the conversation and are there any characteristics attributed to participants?

Reading the results it seemed to be that example quotes were skewed to the Global North over the Global South, particularly in the second part of the results section. Not sure if this was purely because this was best illustrated through these viewpoints, do wonder if there are other examples to rebalance a bit and give voices to those who tend not to get heard? Though acknowledge this is difficult especially as some points are obviously only relevant to particular counties or part of the world.

Line 725 – 736 – The discussion about inclusion of climate-deniers in decision-making by participants was interesting however what was the rationale for not including those who responded in the screening that they did not believe climate change was happening? Obviously there is a tension between potential disruptors and calls that it is not worthwhile engaging with those entrenched in their views however there is also evidence that 'denialists' have opinions on SRM/CDR based on the technology itself e.g. cultural cognition theory or mitigation deterrence.

Figure 2 is not completely clear, what do the authors mean by issue formation for example (it is kind of explained later but long after the reader is first referenced to the figure). Or indeed what 'non-engagement' means?

The discussion section feels like a continuation of the results section as the authors include how they visualise engagement, this mapping would be more appropriate as part of the results section. Their outline of rationales (Fig 3) makes more sense in the discussion section as it outlines the range of points made regarding participation.

The conclusions made are not surprising and their novelty comes from the emerging CDR/SRM technologies and the inclusive workshops.

All REVIEWERS

Formatting of author list	adapted
Nature referencing style	adapted
Methods section at the end of manuscript	adapted
No separate conceptual background section	Integrated in results
No separate conclusion section	adapted
Headings: no punctuation and no third-level headings	adapted

REVIEWER 2

The article examines public preferences and rationales for engagement practices in carbon dioxide removal and solar radiation management. The question addressed is What do diverse publics think about their own role when it comes to decision-making about CDR and SRM technologies? The paper is based on the qualitative analysis of 44 focus groups across 22 countries. It finds various forms of public engagement and rationales for or against participation around CDR and SRM. The article concludes that engagement practices (1) should be adopted to specific methods, (2) reflect on power dynamics, (3) account for publics' previous knowledge and experiences, (4) foster trust in processes and institutions, and (5) value debate and negotiation. The paper's results and conclusions seem isolated and are not particularly novel yet. However, the paper shows promise due to its broad geographical coverage and its contribution to the expansion of the idea of participation applied to CDR and SRM. I recommend the article be accepted pending major revisions, which I detail below:	Thank you very much for this constructive and detailed review. We took your comments and advice thoroughly into account in the revision, and are convinced that this improved the manuscript substantially.
--	--

General -The paper could be strengthened by a more careful reflection on which dimensions are introduced in each section. In the results, you report on the forms and rationales of public engagement, and how the public is delimited (how, why, who) for different places and technologies (where and what). In the discussion, the "when" is introduced and the "where" is omitted. In the conclusion, considerations for meaningful public engagement are introduced. A more careful selection of which dimensions are present in each section, which of their interactions are highlighted and how the argument you construct leads to the introduction of new dimensions would help readers to follow your train of thought.	Based on your comment, we have substantially restructured the results and discussion section. As we describe in greater detail below, the structure of the results section now follows more closely our analytical framework: How, What, Why & Who. We have added a short introduction to the results guiding the reader.
-You use the terms "method" (e.g., line 75), "technology" (e.g., lines 27, 86, 88, "approach" (e.g., lines 17, 116, 179) interchangeably. Clarify which term you use when you refer to both SRM and CDR, which one when you refer to CDR and which one when you talk about specific CDR methods. I suggest using the term "method" for CDR, following the terminology of the IPCC AR6.	Following your suggestion, we have replaced "CDR approaches" by "CDR methods" and now consistently use this terminology throughout the manuscript. We use approaches when we speak of both CDR and SRM. We use technologies and approaches simultaneously when referring to SRM only.
Introduction	
Line 37. Typologies that divide CDR methods into "ecosystems-based" and "engineered" are typically problematic (See e.g., Osaka et al, 2021). It would be more accurate to list the methods without trying to force them into a typology.	We reformulated this sentence and no longer speak of "ecosystem-based" and "technological" sinks.
39. You include the definition of SRM (offsetting global warming by reflecting incoming sunlight). Include the definition of carbon dioxide removal as well.	We reformulated this sentence and now include a short definition of CDR.
40. I recommend nuancing this point. Discussions on the scale and role of the publics in approaches of, for example AR, and the different composite parts of BECCS, are not new. See, for example, https://doi.org/10.1002/wcc.671	We agree that these questions are not ahistorical. We have expanded on this point in para. 5 of the introduction (line 61 ff) where this line of argumentation was already present. We added the suggested reference to the Carton et al. article

	regarding past experiences and controversies with the bioenergy component of BECCS.
78. I suggest mentioning here or elsewhere that certain forms of involvement are given structurally by the methods. For instance, soil carbon enhancement can be done by farmers and other individuals, while industrial capture or underground storage components cannot.	We now expand on this point when discussing the role of technological idiosyncrasies in the results section “what to engage with”.
84. It would probably make sense to describe what you mean by “role” before introducing it in the question. This might clarify how it relates to the “who”, “how”, and “why” dimensions you explore in the paper.	We understand “role” to encompass all the facets that we explore in the paper. Adding them all to the overall research question would make it overly complex. In order to clarify the link to the “how”, and “why” we reformulated the sentence following the research question where we outline our aims.
84. In the results and methods you indicate that the paper explores not only how people see their own role but also that of their fellow citizens. Make sure you are consistent between your RQ and the results your report.	We understand the “role of publics” to be inclusive of “fellow citizens”.
Conceptual background	Please note that to comply with the editorial guidelines of Nature Communications we integrated this section into the Results.
104. This sentence is very long. Consider splitting it into two or more sentences.	We split this – admittedly overly long sentence – into multiple sentences.
107 Mentioning DACCS here sounds odd, because concerns over geological storage and siting do not only concern this method. I suggest referring to geological storage only or to both DACCS and BECCS.	Acknowledging that geological storage matters for various methods, we no longer specifically refer to DACCS here.
136 This table is very useful to guide the reader.	Happy to hear that!
153 To say that “In the field of climate and environment-related decision-making, empirical evidence for such desired, substantive, outcomes of participatory approaches is limited» is quite a bold statement. This will probably depend on how you define «outcome», but I would argue that much of the work of Elinor	We agree that this needs to be nuanced and deserves much broader discussion. With space constraints in mind, we decided to remove this sentence here as we would not have sufficient space to do justice to this important topic.

Ostrom, for example, demonstrates the opposite. I invite you to nuance this statement or include references that support it.	
Research Methods	Please note that in order to comply with the journal's formatting guidelines, we have moved the methods section to the end of the document.
262. Clarify how the recruitment of participants was done, and whether they received any compensation.	As described in the supplementary material, recruitment of participants was organised by a specialized data collection company (NORSTAT), following sampling criteria that we provided. Participants were compensated for their participation in the focus groups. We have added the amounts for each country in the supplementary material.
277. I am not sure that these links to other methods is necessary. The link to DACCS is particularly confusing.	Thanks for bringing this to our attention. In order to simplify, we deleted reference to other methods which are combined in BECCS.
307. It is not clear to me why the analytical category «forms and formats» in table 2 has a different name from the respective approach «on degree or intensity» of table 1 and the section «forms and intensities» in the Results section. Clarify the difference or use the same terminology throughout the text.	We streamlined terminology throughout the results section, now coherently speaking of “forms and intensities” (including in table 3). In our analysis, we first and foremost categorized different forms (e.g. consultation) and corresponding formats (e.g. townhall meetings), interpreting them in the light of “intensities” was a second step. This is why the wording slightly deviates from the one in the table summarizing the literature. In the table summarizing the literature we kept the terminology “degree or intensity”, reflecting its use in the respective literature.
307 Add punctuation marks to the citations. Add year to Chilvers et al.	Added the year and punctuation.
Results	Please note that to comply with the journal's formatting guidelines, we adapted (the formatting of) subheadings throughout this section (e.g. no punctuation is allowed).

-I suggest that you report on the interrelationships between the dimensions of analysis already in this section, rather than (only) keeping them in isolation. For example, by presenting a table where the forms, rationales and constructions of the public are presented together, by moving some of the graphs you present in the discussion to this section, or by exemplifying some of the “ecologies” you observe around different methods in specific regions.

-Given that you have a unique dataset from a diversity of countries, it would be valuable if you could summarize the differences you observed between them. This could provide a more nuanced picture of contexts of participation than generic typologies.

- We have substantially reorganized the results section. Following your advice we have moved the figures and corresponding text from the discussion to the results section. This, we believe, not only makes the interrelation between dimensions more explicit, but also avoid overlaps and redundancies. Thank you for bringing this to our attention.

Regarding the recommendation to summarize differences between countries regarding contexts of participation, we ask the reviewer to consider a compromise.

We have added the need to account for national and cultural modes of civic engagement in our concluding list of recommendations. In our results, we also show how certain political systems and participation cultures might inform rationales for/against public engagement. We give some especially clear examples in which participants spoke to these topics explicitly.

Otherwise, we are wary that this important dimension receives only simplified treatment here. This is for two reasons. Firstly, it is due to our comparative focus on other dimensions and maintaining a more global scope. Secondly, we are particularly wary here of misinterpreting more subtle hints of national/cultural modes of civic engagement, as we are from global North backgrounds. Finally, if we made this central to the paper, there could be 22 different contexts to consider, and simplification (e.g. ‘European’ or ‘African’ modes) would pose its own questions.

We therefore plan to explore country differences in future papers, reducing the number of cases to be representative rather than comprehensive, and relying on additional, secondary data for contextual analyses of political cultures.

	We also acknowledge these limitations in describing our methods: “In reporting the results, we focus on recurring themes that are addressed across various focus groups. Where possible we emphasize context specificities or variances in interpretation of themes across groups. While counts of country mentions offer some indication of the relevance and salience of themes, group dynamics may simply have led to a focus on certain themes and a neglect of others. References to country mentions and counts thus need to be interpreted with caution. Given the multi-technology, multi-country approach of this study, however, in-depth discussions of individual technologies and countries cannot be provided in this article.”
-In my opinion, there are too many quotes in the text and some of them are too long for the points you want to exemplify. I invite you to select only the quotes you find most insightful and, when possible, keep them rather short.	Given the empirical novelty of our rich dataset, we consider it important to provide space for participants’ reflections in their own words. We also see this as an opportunity to give visibility to voices from countries which have been largely underrepresented so far in public perceptions studies. Reading the manuscript with some distance, we agree, however, that we can be more selective on which ones to present. We, thus, deleted some and shortened several quotes to make them more targeted to the arguments presented, while still providing sufficient context for interpretation.
374 You write that “Narratives about self-engagement reflect efforts of assuming agency and taking responsibility”. Is this part of your interpretation?	We deleted this sentence as we realized it was repetitive. This interpretation is strongly supported by our data which show that publics’ reflections on their own role in many cases are tied to wider narrations of how they themselves try to contribute to tackling the climate crisis, assuming responsibility for changing consumption practices for example.

392 In line with the other approaches, clarify how you defined consultation approaches.	We added our definition in the first paragraph on consultations (line 309).
419. Replace DAC by DACCS. DAC is not CDR.	Thanks for spotting this, we corrected it.
485. I find this dialogue confusing. Perhaps it is possible to omit the parts that do not add to the argument	We now only show the elements of the discussion directly speaking to the theme of the section (protests). In order to enhance readability, we have, furthermore, adapted speaker “names”, now referring to “Respondent 1”, “Respondent 2”, “Moderator”.
567 In some cases, it is not clear to me what you mean by “in general” and “mentioned for all” in the table. For example: “Afforestation and restoration, Soil carbon sequestration and biochar, CDR in general; mentioned for all”.	Thanks – we realized that this wording was not clear. We now clarify in a Table Note that the “in general” is used when participants speak of CDR or SRM as a whole and do not differentiate or mention a specific method/approach. We removed references to “mentioned for all” in order to better highlight where the focus lies.
570. This subtitle should probably refer to both the rationales and the “who”.	You are right that this section is both on the “why” and the “who”, we have adapted the subheading accordingly.
570 The analytical lens you use to build the rationales is not clear to me. It seems as if they were rather organized according to the different types of participation (with more or less empowerment) than according to, for instance, the rationales of Fiorino, 1989.	Guided by our awareness of definitions of participation according to motivation/rationale, the specific categories for rationales were inductively developed, based on how participants reflect on why they consider engagement desirable (or not), i.e. during the qualitative coding of transcripts all segments where participants state the reason of why/why not they consider engagement desirable were identified. In a further analytical step, it became clear that many of these rationales can be mapped onto /interpreted in light of the broader analytical dimensions of Fiorino as they reflect instrumental, normative or substantive reasonings: Participants reasonings for public engagement around increasing acceptance for certain methods or reducing opposition towards them

	reflect instrumental rationales. Reasonings around how specific local knowledge can improve decisions about implementation and reduce potential trade-offs reflect substantive rationales. Reasoning around the idea that those affected should have a right to express their opinions reflect normative rationales. In presenting our results, we decided in a first step to stay close to the data as this offers greater nuance and allows us to empirically carve out the broader categories by Fiorino for the context of CDR and SRM. In the discussion section, we offer an interpretation in light of Fiorino’s typology.
790. Some of the contents of this table seem to be absent in the text or lack an explicit connection (e.g., the rationale "other priorities of publics", the constructions of transformative and resisting publics). Make sure to maintain consistency between the interpretations in the text and the categories in the table. Speak in the first person when describing your interpretations. So far, the authors seem to be absent in the text.	The text focuses on the most discussed rationales, but also mentions “other” rationales less often mentioned (e.g. line 703). We have reorganized table 4 in order to mirror the structure of the main text (rationales for engagement first, followed by rationales against). Furthermore, we have simplified table 4 and removed the column on conception of the public. We now introduce this additional element in figure 4 which – following your advice – we moved to the results section. We now also speak in the first person, signaling our role as those interpreting results, for example: “We suggest that different rationales are... “ We have adopted more active language in various other parts, for example in line 480: “We interpret these narratives as a reflection of participants’ efforts of maintaining agency in the context of complex and multi-layered problems”
Discussion	
-The diversity of countries is a key feature of your paper but is largely absent from this section. It would be useful, for example, to discuss how the culture and history of countries, in general terms, determine the	In the discussion section, we have added a point on the need to account for national contexts and modes of civic engagement. Drawing more detailed conclusions on the impact of cultural and historical contexts on

forms of participation mentioned by participants.	engagement preferences would, however, require deeper country-specific analyses that rely on additional, secondary data. We would also stress again that given the multi-technology, multi-country approach of this study in-depth discussions of individual countries cannot be provided in this manuscript. These deserve deeper dives elsewhere.
-Although you describe your results as contributing to the "expanded role of deliberative thinking" and highlight the importance of "ecosystems of participation", the relationship of your results to these concepts would benefit from a richer discussion.	We are aware of very few studies on public engagement and participation that take the perspectives of publics themselves (rather than how "experts" think about the roles of publics) in general, and even less so applied to the context of CDR and/or SRM. We therefore believe our claim about expanding knowledge on deliberative thinking is accurate and stands. We have, furthermore, added an "key consideration" in the discussion section ("Recognizing that participation exists across a spectrum or ecology of subjects, objects, and models"), now explicitly discussing the relation of our results to ecologies of participation.
803. Clarify in more detail how your results contribute to the "expanded role of deliberative thinking", in comparison to "conventional" ideas of public engagement and existing systemic mappings of participation, like the ones of Chilvers et al, 2021 and Chilvers et al, 2023. How does this represent a new way of thinking about participation for decision makers, citizens and/or social scientists working on CDR/SRM?	Addressed above.
812. The dynamics of participation over time is an interesting point to mention! You can return to the work of Stauffacher et al. (2008) here. Much of the information here is new and could therefore be included in the results.	Following your suggestion, we included this in the results section and return to Stauffacher et al. in this context.
825. Some examples of the ways of participation mentioned by the publics that are not traditionally captured in participation theories would be helpful here.	The forms of "self-engagement" (e.g. in afforestation practices) that we mention in the sentences that follow illustrate this point. We have slightly reformulated this and added "for example" to make the link clearer.

826. Do you refer to “ecologies” of participation?	Indeed, thanks for pointing this out. We corrected to “ecologies”.
830. The role of the citizens, how they see themselves interacting with different types of technologies and why could already be brought up in the results.	Following your suggestion, we included this in the results section.
839 As previously said, I suggest moving these figures and text to the results, if the “when” is part of the role of the publics you are exploring.	We have integrated this in the results section. We consider this as part of the “what” in our analytical framework, i.e. what is the object to be engaged with (e.g. is it decision-making with regard to a specific method, is it implementation of a specific method?). In order to make this clearer to the reader from the outset, we have reformulated the description of this analytical dimensions in our table in the methods (table 2).
839 On the Figure II, B: 1) Why is the axis called self-engagement? Is it restricted to the category of “self- and community engagement”? 2) It is not clear to me why you have two dots for soils and AF/RF (on issue formation and implementation) and only one dot for the other methods. Does the dot for “All” exclude AF/RF? 3) The graph suggests that for methods like BECCS, DACCS and EW the most discussed type of engagement is non-engagement. An expansion of this on the text would be relevant. 4) Clarify how the intensity of discussion is measured for this figure.	1) Thanks to your comment we realized that the labels of the vertical axis were misleading and inconsistent. We have reworked the vertical axis of the figure and now refer to “top down “ (i.e. institution-led) and “bottom-up” (i.e. citizen-led”) practices . 2) We specify in the figure notes that the categories are not mutually exclusive (i.e. the dot for “all” – more top-down provision of information is asked for all approaches – does not exclude bottom-up engagement in sharing experience with and knowledge about AF/RF). This means that several forms of engagement might be considered relevant for a specific method. For example, AF and SOILS are frequently discussed in relation to self-engagement in information sharing, but also with regard to practical implementation. 3) As we now state more explicitly in the text accompanying the figure DACCS/BECCS/EW, were mostly addressed regarding consultation processes in site-specific and siting-related questions, predominantly referring to top-own, invited, forms of participation. 4) We clarify now in the figure Note that the mapping is based on our interpretation

	of results (notably the technological foci identified in table 3).
882 As mentioned before, this might also be given structurally by the methods.	See our response further above.
882 Do you observe in your material that public agency and technical simplicity are coupled or is this your interpretation?	This is our interpretation of the empirical data. Following your advice for using more “active” language, we have reformulated the statement as follows: “We can infer that conceptions of public agency are intertwined with perceptions of technical simplicity, adaptability, and applicability.”
898 I encourage you to elaborate on the participants' scepticism about public engagement. What is the reason for this discrepancy among participants? What does this tell us about the design of participatory processes for CDR / SRM?	We have tried to reflect this as best as we can within the paper, already noting the key considerations “adopting power-sensitive practices”, and “establishing trust and procedural legitimacy” which the point you raise. We offer further interpretations regarding varying roots of skepticism in the newly added point on “accounting for national contexts”. Despite being highly relevant, space constraints do not allow us to develop this aspect any further.
Conclusions	Please note that the formatting guidelines of Nature Communications do not allow for a separate conclusion section – hence the restructuration.
-The conclusions do not sound very novel yet. However, the material offers such rich sources, that there must be much more nuanced and specific that can be said. Some of these points (e.g., on power and agency) and how they shape dynamics of participation across methods could be taken up more strongly in the discussion.	As described above we added two new paragraphs, (one on country contexts) in the discussion section. Overall, we would like to highlight that our manuscript is first and foremost offering empirical novelty. We are not aware of any other studies providing first-hand empirical insights into how publics in varied geographies see their own roles in regard to the governance of CDR and SRM approaches.
Discuss in light of what is new here, how it builds on existing literature, how it expands the concept of ecologies of participation or public engagement and links it to CDR and SRM.	We have reworked the discussion section and have added two new paragraphs, one summarizing the contribution to participation scholarship and the other one highlighting the need for nuancing country contexts.
Appendix -If possible, include in the preparatory reading material provided to participants.	We have included the preparatory material in the supplementary information.

REVIEWER 3

Overview: This paper provided decent insights into participation of publics across 22 countries in both the Global North and the Global South in considering governance of climate intervention technologies. The introduction explores the literature to help contextualise interpretations of what ‘participation’ looks like. The rationale of the paper is sound and the framework developed is clear but it is hard to see how this could be replicated. The methods left a lot of questions for the reader even taking into consideration the supplementary information. Participants were given a week to read information given to them on a range of different techniques across both SRM/CDR but it was not clear if they had engaged with the material or done their own research especially considering the range of techniques discussed over the two hour session.	Thank you very much for the positive feedback, and the very helpful comments on our manuscript. We have thoroughly revised the manuscript along your suggestions and included more detailed information on data collection in the supplementary information.
Introduction: Overall a thoughtful overview of participation of publics in the emerging technologies space.	Happy to hear that!
Methods: I acknowledge there are 22 groups in the sample but two hours seems like an incredibly short time for a workshop discussing complex and novel information for participants. The authors describe the content of the information in the appendix. Could they include the actual material content rather than the description provided of what it contained? Is there any evidence that describing what the authors have termed ‘biogenic’ approaches as those ‘that change how we use nature’ changes how people will respond technologies by not using ‘nature-based’? In my mind the updated phrasing seems similar to a ‘messing with nature’ framing. In addition the word nature itself	Please note that in order to comply with the journal’s formatting guidelines, we have moved the methods section to the end of the document. To be clear, there are 44 groups in 22 countries. We decided on the 2 hours given the resources and budget for the project and the risk of fatigue. In deliberating pros and cons of different formats and taking our decision, we also built on experience from other focus groups on CDR and SRM of similar length which are aimed at “opening up” largely unfamiliar topics and initiating discussion and reflection. For example Asayama et al. 2017 : around 2 hours; Carvalho and Riquito : 1 to 2 hours; MacNaghten and Szerszynski : around 3 hours; Wibeck et al. 2015 : 40 to 90

can mean many different things to different people and cultures.
Were the translated (and original) materials piloted before they were used in the actual workshops? (Based on the previous comment that terminology and phrases can be interpreted differently between countries and cultures so hard to know how comparable different strategies may be).
In the description of the material it says the pros and cons were abbreviated as much as possible to limit framing, it is not clear what this information was or how it was abbreviated?

minutes; Wibeck et al. 2017 : 1 to 2 hours.
(*References at the end of this cell)

We are also aware of studies published in Nature journals that have even shorter focus groups, such as <https://www.nature.com/articles/s41560-023-01196-w> (90 min focus groups), <https://www.nature.com/articles/s41415-023-6052-x> (60 minutes) and <https://www.nature.com/articles/s41599-024-02661-x> (90 minutes).

Following your suggestion, we have included the preparatory reading material in the supplementary information. Participants were encouraged to discuss the materials with family and friends beforehand.

We acknowledge some evidence for framing effects in the literature. In preparing the material we aimed for “simple” language that can easily be understood by non-specialists. Sometimes, as in the case of the formulation “that change how we use nature” required a compromise with “precision”. While we cannot exclude some degree of framing, we observe similar results and preferences for “ecosystem”-based approaches in a large-n, representative, quantitative survey that was coupled with our focus groups. We are, thus, confident that our simplified language did not introduce additional biases. In any case, perceptions of risks and benefits of the respective approaches are not at the center of this manuscript.

Feedback by colleagues and people not familiar with the topic (including the teams of moderators) was sought during the preparation process. Native-language moderators (12 languages) were briefed on the materials and asked to provide feedback and communicate aspects that

	were not clear to them. This written feedback was discussed in a joint call between our team at Aarhus University and NORSTAT and taken into account in the finalization of the materials – along with in-house checks by native-language individuals also being done at Norstat. All technical terms, compiled in a list of around 25 keywords, were translated from English into their native languages by academic experts (all colleagues in climate and energy governance known to the authors) and previously used in a large-n survey that accompanied the focus groups. Regarding your last point, the preparatory reading material – now included in the supplementary information – shows how pros and cons were introduced to focus group participants. *Asayama, S., Sugiyama, M., & Ishii, A. (2017). Ambivalent climate of opinions: Tensions and dilemmas in understanding geoengineering experimentation. Geoforum, 80, 82–92. https://doi.org/10.1016/j.geoforum.2017.01.012 Carvalho, A., & Riquito, M. (2022). ‘It’s just a Band-Aid!’: Public engagement with geoengineering and the politics of the climate crisis. Public Understanding of Science, 096366252210953. https://doi.org/10.1177/09636625221095353 Macnaghten, P., & Szerszynski, B. (2013). Living the global social experiment: An analysis of public discourse on solar radiation management and its implications for governance. Global Environmental Change, 23(2), 465–474. https://doi.org/10.1016/j.gloenvcha.2012.12.008 Wibeck, V., Hansson, A., & Anshelm, J. (2015). Questioning the technological fix to climate change – Lay sense-making of geoengineering in Sweden. Energy Research & Social Science, 7, 23–30. https://doi.org/10.1016/j.erss.2015.03.001 Wibeck, V., Hansson, A., Anshelm, J., Asayama, S., Dilling, L., Feetham, P. M., Hauser, R., Ishii, A., & Sugiyama, M. (2017). Making sense of climate engineering: A focus group study of lay publics in four countries. Climatic Change, 145(1), 1–14. https://doi.org/10.1007/s10584-017-2067-0
Results:	Please note that to comply with the journal’s formatting guidelines, numbering of headings and third-level headings had to

	be removed. We also had to slightly adapt the headings to avoid punctuation.
Line 485 – 497 – how does the reader distinguish between respondents and/or moderators in the conversation and are there any characteristics attributed to participants?	In order to enhance clarity and readability, we have changed speaker names to “respondent 1”, “respondent 2”, “moderator” in conversations involving multiple speakers, thus allowing to easily distinguish between respondents and moderators. While we have information on the overall composition of each group, we do not have information on the socio-demographics or other characteristics of each speaker. Alongside removing names and references to where participants live, these had to be removed during the transcription process in order to comply with data protection standards.
Reading the results it seemed to be that example quotes were skewed to the Global North over the Global South, particularly in the second part of the results section. Not sure if this was purely because this was best illustrated through these viewpoints, do wonder if there are other examples to rebalance a bit and give voices to those who tend not to get heard? Though acknowledge this is difficult especially as some points are obviously only relevant to particular countries or part of the world.	Thank you for raising this important point. We have taken this into account in the revision (see for example the newly added quote from a Saudi Arabian group in rationales against active public engagement), while balancing the requests of another reviewer to reduce the overall number of direct quotes. As you rightly point out, in the selection of quotes we also had to consider that not all groups speak equally to a particular theme. This, however, does not compromise novelty. To our knowledge, also voices from Poland or Spain have not received a lot of attention in previous work.
Line 725 – 736 – The discussion about inclusion of climate-deniers in decision-making by participants was interesting however what was the rationale for not including those who responded in the screening that they did not believe climate change was happening? Obviously there is a tension between potential disruptors and calls that it is not worthwhile engaging with those entrenched in their views however there is also evidence that ‘denialists’ have opinions on SRM/CDR based on the	As you rightly mention this decision was motivated by the aim of ensuring some common ground among participants, thus allowing for constructive discussions of the methods in question – as well as the fact that the approaches are, to an extent, motivated by the issue of climate change. We therefore deemed it sensible to not focus, for this step, on those not believing that climate change was occurring (which, in a 30 country survey exercise accompanying our focus groups, amounted

technology itself e.g. cultural cognition theory or mitigation deterrence.	to 4.6% of the total sample, as little as 1-2% (in Indonesia, India, and Nigeria) and no more than 11% (in the United States). As such, this was not a large set of the population, particularly outside of the United States. Having an individual with such beliefs, in a focus group of 6-8, would therefore be unrepresentative of national populations. Given that methodological literature on focus groups suggests some degree of homogeneity in group composition, we consider this decision legitimate. However, we agree that future research engaging with these particular groups would yield interesting additional insights, particularly on risks of mitigation deterrence.
Figure 2 is not completely clear, what do the authors mean by issue formation for example (it is kind of explained later but long after the reader is first referenced to the figure). Or indeed what 'non-engagement' means?	These dimensions are adapted from Chilvers et al. 2021 where issue formation broadly refers to "practices that emphasizes the expression of views". We have added an explanation to the figure legend. Thanks to your comment we realized that the labels of the vertical axis were misleading and inconsistent. We have reworked the vertical axis of the figure and now refer to "top down " (i.e. institution-led) and "bottom-up" (i.e. citizen-led") practices.
The discussion section feels like a continuation of the results section as the authors include how they visualise engagement, this mapping would be more appropriate as part of the results section. Their outline of rationales (Fig 3) makes more sense in the discussion section as it outlines the range of points made regarding participation.	Following your suggestions, we have substantially restructured the Results and Discussion sections, integrating amongst other things the mentioned visualizations into the results section. After careful consideration and in line with another reviewer's suggestion we decided to also include the figure mapping rationales for /against engagement into the results section. This allowed us to avoid overlaps between the two sections.
The conclusions made are not surprising and their novelty comes from the emerging CDR/SRM technologies and the inclusive workshops.	We agree and are convinced that our manuscript offers first and foremost empirical novelty. We are not aware of any other studies providing first-hand empirical insights into how publics in varied geographies see their own roles in regard to

	the governance of CDR and SRM approaches. We have furthermore added two new paragraphs in the Discussion, summarizing the contribution to participation scholarship as well as highlighting the need for nuancing country contexts. Please note that the formatting guidelines of Nature Communications do not allow for a separate conclusion section – hence the restructuration.
--	---

Reviewers' Comments:

Reviewer #2:

Remarks to the Author:

I am satisfied with the authors' responses to my comments and feel that the manuscript is much improved. I particularly welcome the greater coherence between sections, the better connection of the discussion/conclusion with the narrative of the rest of the paper and the reformulation of the discussion/conclusion, which now makes an interesting reflection on conditions for better participation across geographies.

There is one particular issue that still requires attention. I do not agree with the authors' decision to move the "conceptualizing public engagement" section, typologies and table 1 to the results section. Although they argue that this is to comply with the Nature Communications editorial guidelines, it is too long compared to similar articles. A brief recapitulation of their methods and conceptual framework would be fine to guide the readers, but in its present form, the role of the text goes far beyond that. I recommend the authors to integrate this text into the introduction.

Reviewer #3:

Remarks to the Author:

The authors have acted upon my queries in their revised manuscript and the restructuring has much improved the flow. The exploration of participation in a range of countries on this topic is underexplored and this provides a good examination of the issue which will be of wider interest.

REVIEWER 2

I am satisfied with the authors' responses to my comments and feel that the manuscript is much improved. I particularly welcome the greater coherence between sections, the better connection of the discussion/conclusion with the narrative of the rest of the paper and the reformulation of the discussion/conclusion, which now makes an interesting reflection on conditions for better participation across geographies.	We highly appreciate your rigorous and constructive engagement with our manuscript, and thank you for all the helpful suggestions!
There is one particular issue that still requires attention. I do not agree with the authors' decision to move the "conceptualizing public engagement" section, typologies and table 1 to the results section. Although they argue that this is to comply with the Nature Communications editorial guidelines, it is too long compared to similar articles. A brief recapitulation of their methods and conceptual framework would be fine to guide the readers, but in its present form, the role of the text goes far beyond that. I recommend the authors to integrate this text into the introduction.	Thank you for raising this point. To address your point, we have integrated a part of this section into the introduction as well as slightly shortened the section. We are afraid that integrating the entire section into the introduction would be incompatible with the editorial guidelines concerning structure and length of the introduction. Furthermore, sub-headings are not allowed in the introduction which would impede readability. The solution that we have chosen has previously been implemented in other social science papers in Nature Communications, for example: Svoboda et al. 2022: https://doi.org/10.1038/s41467-022-35391-2

REVIEWER 3

The authors have acted upon my queries in their revised manuscript and the restructuring has much improved the flow. The exploration of	Thank you very much for taking the time to rigorously engage with our manuscript!
--	--

participation in a range of countries on this topic is underexplored and this provides a good examination of the issue which will be of wider interest.	
---	--